# How reliable are process-based $^{222}$radon emission maps? Results from an atmospheric $^{222}$radon inversion in Europe

Fabian Maier[1], Eva Falge[2], Maksym Gachkivskyi[3], Stephan Henne[4], Ute Karstens[5], Dafina Kikaj[6], Ingeborg Levin[3,†], Alistair Manning[7], Christian Rödenbeck[1], and Christoph Gerbig[1]

[1]Department of Biogeochemical Signals, Max Planck Institute for Biogeochemistry, Jena, Germany
[2]Zentrum für Agrarmeteorologische Forschung, Deutscher Wetterdienst, Braunschweig, Germany
[3]Institut für Umweltphysik, Heidelberg University, Heidelberg, Germany
[4]Empa, Laboratory for Air Pollution / Environmental Technology, Dübendorf, Switzerland
[5]ICOS Carbon Portal, Department of Physical Geography and Ecosystem Science, Lund University, Lund, Sweden
[6]National Physical Laboratory, Teddington, UK
[7]Met Office Hadley Centre, Exeter, UK
[†]deceased, 10 February 2024

*Correspondence to*: Fabian Maier (fmaier@bgc-jena.mpg.de)

**Abstract.** The radioactive noble gas radon ($^{222}$Rn) is a suitable tracer for atmospheric transport and mixing processes that can be used to evaluate and calibrate atmospheric transport models or to estimate greenhouse gas emissions using the so-called Radon-Tracer method. However, these applications require reliable estimates of the $^{222}$Rn fluxes from the soil. This study evaluates two process-based $^{222}$Rn flux maps in central Europe in 2021 using the flux results from a one-year $^{222}$Rn inversion. The maps are based on different soil moisture reanalysis products (GLDAS-Noah and ERA5-Land), which are used to describe the diffusive $^{222}$Rn transport in the soil. The $^{222}$Rn inversion was conducted using the CarboScope-Regional inversion system, and observational data from 17 atmospheric sites in central Europe in 2021. We observe that, in particular, the ERA5-Land based $^{222}$Rn flux map underestimates the data-driven fluxes from the inversion. Our inversion yields ca. 20% (GLDAS-Noah) to almost 100% (ERA5-Land) larger $^{222}$Rn fluxes than the respective process-based prior fluxes within a domain covering Germany. Also, the temporal variability seems to be underestimated by the process-based flux maps. Using a flat (uniform) prior inversion, we found a significant anti-correlation of -0.6 (and -0.8) between the posterior $^{222}$Rn flux and the GLDAS-Noah (and ERA5-Land) soil moisture time series, indicating that soil moisture is an important driver for the temporal variability of the $^{222}$Rn fluxes. To investigate the robustness of our flux estimates, we run the inversion with three different transport models (STILT, FLEXPART, NAME). The respective annual mean flux results agree within ca. 10%.

## 1 Introduction

Inverse modelling (e.g., Newsam and Enting, 1988) is a well-established method for constraining surface fluxes of greenhouse gases (GHGs) by minimising the mismatch between observed and simulated atmospheric dry air mole fractions of the respective GHG. However, limitations in atmospheric transport models, such as inadequate description of vertical mixing within the planetary boundary layer (PBL), can lead to systematic biases in such top-down flux estimates (e.g., Schuh et al., 2019; Schuh et al., 2022; Munassar et al., 2023). Therefore, careful quantification of transport model uncertainties is essential for a reliable flux estimation. As this is far from straightforward, potential systematic biases in atmospheric transport models are often assessed by using an ensemble of different transport models in the inversion framework to investigate their impact on the flux estimates (e.g., Geels et al., 2007; Peylin et al., 2013; Monteil et al., 2020; Schuh et al., 2022) or by employing computationally expensive ensemble methods (e.g., Steiner et al. 2024).

A more direct way of quantifying transport model uncertainties is to compare modelled and measured atmospheric activity concentrations of the radioactive noble gas radon ($^{222}$Rn), which is the first gaseous component in the decay chain of uranium that escapes from the soil into the atmosphere (Karstens et al., 2015). As its lifetime (3.8 days) is comparable to the ventilation time scale of the PBL, atmospheric $^{222}$Rn observations over land contain suitable information on vertical mixing (Jacob and Prather, 1990) and can be used to validate (and possibly even improve) the performance of atmospheric transport models in this respect (e.g., Jacob and Prather, 1990; Chevillard et al., 2002; Gupta et al., 2004; Galmarini, 2006; Zhang et al., 2008; Arnold et al., 2010; Zhang et al., 2021). However, this requires accurate $^{222}$Rn flux maps that are suitable for modelling atmospheric $^{222}$Rn activity concentrations.

There are various global and regional $^{222}$Rn flux maps available, which differ in the methods and the complexity used to describe the $^{222}$Rn exhalation from the soil (e.g., Rasch et al., 2000; Conen and Robertson, 2002; Zhou et al., 2008; Szegvary et al., 2009; Griffiths et al., 2010; López-Coto et al., 2013; Karstens et al., 2015; Karstens and Levin, 2024). In an extensive study, Karstens et al. (2015) developed two process-based $^{222}$Rn flux maps for Europe and compared them with existing $^{222}$Rn flux maps from the literature. Their study revealed large spatio-temporal differences between the different $^{222}$Rn flux maps, which can be on the order of the $^{222}$Rn fluxes themselves, and illustrates the substantial uncertainty associated with $^{222}$Rn flux maps.

In principle, continuous $^{222}$Rn flux measurements can be used to validate (and calibrate) the $^{222}$Rn flux maps (Griffiths et al., 2010; Manohar et al., 2013; Karstens et al., 2015). However, such measurements are sparse and typically only representative for very local spatial scales and often contradict large-scale flux estimates due to the high degree of the soil parameter

inhomogeneities. In a technical report on their [222]Rn flux model, Karstens and Levin (2024) compared the modelled [222]Rn flux with continuous [222]Rn flux measurements from one site during six months and showed that it is difficult to validate the [222]Rn flux map of an entire continent with a sparse set of flux measurements. Therefore, long-term, continuous and reliable [222]Rn flux measurements with a high spatial representativeness across different parts of Europe are needed for further validation

studies. For this reason, Karstens and Levin (2024) recommended performing a [222]Rn inversion to evaluate the [222]Rn flux maps, which is a more representative approach as the atmosphere integrates fluxes from larger areas. In our study, we implemented their suggestion and investigated whether we could evaluate the quality of [222]Rn flux maps using inverse modelling.

However, there is a conflict in performing a [222]Rn inversion, since it intrinsically assumes that atmospheric transport is well

known and without systematic biases (Schuh et al., 2019). As this is not the case (see above), there is a risk that the inversion will adjust the [222]Rn fluxes to compensate for unknown biases in the transport model (especially if the transport model uncertainties are not correctly described in the inversion framework). For example, if the vertical mixing in the transport model were too strong, the model would underestimate the [222]Rn activity concentrations, even if the [222]Rn fluxes are correct. As a consequence, the inversion would falsely increase the [222]Rn fluxes. Thus, such incorrectly adjusted [222]Rn fluxes are useless for

modelling [222]Rn concentrations and validating other transport models.

Therefore, we use an ensemble of transport models and an ensemble of a priori flux maps in our inversion system to carefully quantify the impact of potential biases in the transport model and in the prior fluxes on the [222]Rn inversion results. These transport models differ, for example, in the parameterization of turbulent motion and convection, and in the underlying

meteorological data (see Sect. 2.3). Thus, by analyzing the influence of the transport models on the [222]Rn flux estimates, we can assess the robustness of our inversion results. Furthermore, we only use afternoon observations (or nighttime observations for mountain sites), when the atmosphere is typically well-mixed and the models are expected to perform best (Gerbig et al., 2008). By doing so, we aim to obtain reliable [222]Rn flux estimates that can be used to evaluate process-based [222]Rn flux maps.

In our study, we evaluate the two process-based [222]Rn flux maps from Karstens and Levin (2022a,b), which have been updated and further refined within the framework of the traceRadon project (Röttger et al., 2021). The maps are based on two different soil moisture products used to describe the [222]Rn transport through the soil (see Sect. 2.1). We investigate if we can use our [222]Rn inversion results to assess which of the two [222]Rn flux maps is best suited for a domain in central Europe well covered by [222]Rn observations, and if we can derive realistic estimates of their uncertainties (Sect. 3.2). Furthermore, we want to

investigate whether we can learn something about the [222]Rn exhalation process from the inversion results, e.g. what information about soil moisture variability is contained in the [222]Rn observations (Sect. 3.3). In this context, we will also try to improve the process-based [222]Rn flux maps by using different soil moisture data (Sect. 3.4).

## 2 Methods

Figure 1 provides an overview sketch of the $^{222}$Rn inversion process. The individual compartments are described in the following sub-sections.

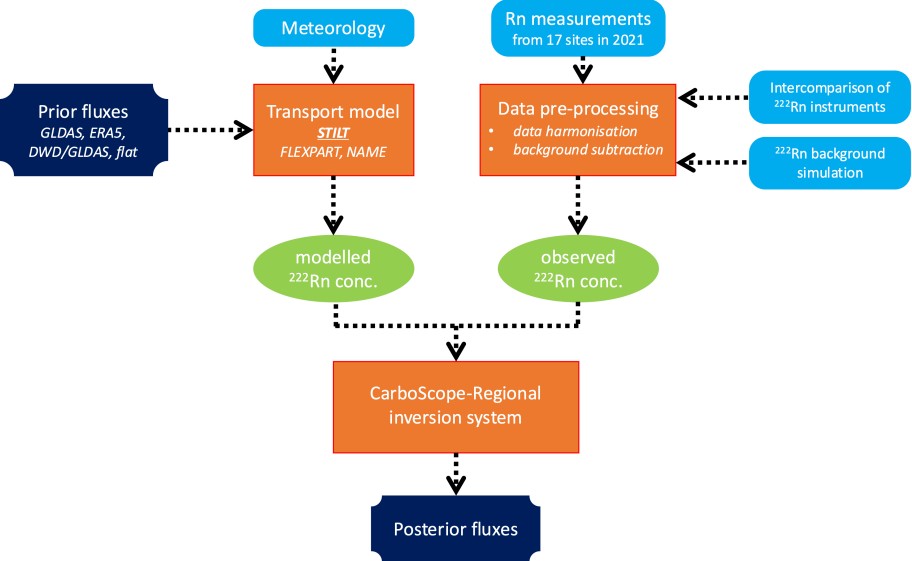

**Figure 1: Sketch of the inversion process.**

### 2.1 Process-based $^{222}$Rn flux maps

In this study, we investigate different process-based $^{222}$Rn flux maps, all of which are based on the $^{222}$Rn flux model developed by Karstens et al. (2015) for an "infinitely deep unsaturated homogeneous soil". We will mainly focus on the $^{222}$Rn flux maps from Karstens and Levin (2022a,b), which are based on GLDAS-Noah and ERA5-Land soil moisture and porosity data to describe the $^{222}$Rn transport through soil. However, we also want to analyze a third, alternative $^{222}$Rn flux map that we have prepared for this study using high-resolution soil moisture and porosity data from the German Weather Service (DWD, see Tab. 1). In the following, we briefly describe this $^{222}$Rn flux model, but the reader is referred to Karstens et al. (2015) and Karstens and Levin (2024) for more comprehensive details.

The main assumption of the $^{222}$Rn model is that $^{222}$Rn exhalation from the soil occurs mainly by diffusive transport (Nazaroff, 1992). This assumption is justified because the $^{222}$Rn activity concentration in the air in the soil is several orders of magnitude higher than in the ambient air above the soil (Čeliković et al., 2022). Assuming steady state conditions and a $^{222}$Rn source $Q$ that is constant with depth, the following equation can be derived for the $^{222}$Rn flux $j$ at the soil surface ($z = 0$)

$$j(z = 0) = -Q\sqrt{\frac{D_e}{\lambda}}\tanh\left(z_G\sqrt{\frac{\lambda}{D_e}}\right) = -Q\,\overline{z}\,\tanh\left(\frac{z_G}{\overline{z}}\right). \tag{1}$$

Hence, the flux $j$ (in Bq m$^{-2}$ s$^{-1}$) depends on the $^{222}$Rn source $Q$ (in Bq m$^{-3}$ s$^{-1}$), the effective diffusion coefficient $D_e$ (in m$^2$ s$^{-1}$),

the $^{222}$Rn decay constant $\lambda$ (in s$^{-1}$) and the water-table depth $z_G$ (in m). The $^{222}$Rn relaxation depth $\overline{z}$ (in m) is given by $\overline{z} = \sqrt{D_e/\lambda}$. The $^{222}$Rn source $Q$ can be described as a product of the concentration of radium ($^{226}$Ra, i.e. the precursor of $^{222}$Rn) in the soil, the $^{222}$Rn decay constant, the dry bulk density of the soil, as well as the emanation coefficient, which describes the probability that the $^{222}$Rn atoms can escape from the soil grains, in which they were formed, into the soil air. According to the parameterization from Zhou et al. (2008), this emanation coefficient depends on the soil texture, soil moisture, soil porosity

and soil temperature. The $^{226}$Ra concentration was calculated using a map of the uranium ($^{238}$U) content in the soil from the European Atlas for Natural Radiation (EANR; Cinelli et al., 2019) and assuming secular equilibrium between $^{238}$U and its daughter $^{226}$Ra (see Karstens and Levin, 2024).

The effective diffusion coefficient $D_e$ is described with the parameterization from Millington and Quirk (1960), which depends

on soil porosity and soil moisture. Karstens and Levin (2024) used the average soil moisture and porosity data from the upper 40 cm of the soil to calculate the diffusion coefficient (and the emanation coefficient). Hence, they assumed that the $^{222}$Rn released from the soil surface originates mainly from the top 40 cm of the soil. In Sect. 3.4, we revisit this assumption. The $^{222}$Rn flux model takes also into account a temperature dependence of the diffusion coefficient according to the parameterization from Schery and Wasiolek (1998).

Furthermore, the $^{222}$Rn flux from the soil is reduced in regions with shallow water-table depths as the soil water hinders the $^{222}$Rn exhalation. It is described by the factor $\tanh\left(\frac{z_G}{\overline{z}}\right)$ in Eq. 1. For water-table depths $z_G$ that are large compared to the $^{222}$Rn relaxation depth $\overline{z}$ (i.e., $z_G >> \overline{z}$), this factor becomes 1. All three $^{222}$Rn flux maps used in this study were calculated using Eq. 1 but with different soil moisture data products as listed in Tab. 1. The two $^{222}$Rn flux maps from Karstens and Levin (2022a,b),

based on GLDAS-Noah and ERA5-Land soil moisture respectively, cover a slightly smaller area compared to our model domain. Therefore, we have spatially extended these $^{222}$Rn flux maps by filling the extended (land) areas with the respective daily mean land fluxes of the whole domain (see Fig. 3a,b). In the following, we refer to these two $^{222}$Rn flux maps as "GLDAS" and "ERA5", respectively.

For the third $^{222}$Rn flux map, we make use of the high-resolution soil moisture data from the AMBAV (AgrarMeteorologische Berechnung der Aktuellen Verdunstung, Herbst et al., 2021) model, which is a water balance model operated by the German Weather Service (DWD) for agricultural purposes that provides daily, 1 km$^2$ resolution soil moisture data for Germany. Thereby, information on regional soils from detailed geological maps (Hartmann et al., 2024) is incorporated into the model. The AMBAV model calculates soil moisture data for crop and grass-covered soil types. Whilst the GLDAS-Noah and ERA5-

Land models assume that the soil porosity is constant over the entire soil column, AMBAV uses vertically resolved soil porosity data (from Hartmann et al., 2024). We combined the AMBAV soil moisture product with soil moisture data for forests

simulated with the forest hydrological model LWF-Brook90 (Hammel and Kennel, 2001), to obtain a high-resolution soil moisture map for the dominant land cover types in Germany, which we refer to as "DWD" soil moisture hereinafter. The basis for the classification of the pixels into arable land, grassland and forest is the land cover data from CORINE Landcover (2018), the classification of the forest pixels into the main tree species (beech, oak, spruce and pine) was carried out using the tree species map by Blickensdörfer et al. (2022). To be consistent with Karstens and Levin (2024), we initially use the average DWD soil moisture (and porosity) of the top 40 cm of the soil. However, in Sect. 3.4, we further investigate how the vertical averaging of the soil moisture impacts the $^{222}$Rn fluxes. Since the DWD soil moisture is only available for Germany, we fill the area outside Germany (and the few gaps over German cities) with the GLDAS-Noah soil moisture data so that we can calculate a complete $^{222}$Rn flux map for Europe. To be consistent, we also apply the porosity data used by the GLDAS-Noah model for areas outside Germany. We call this third $^{222}$Rn flux map "DWD/GLDAS". All three $^{222}$Rn flux maps are re-gridded to a horizontal resolution of 0.05° x 0.05°.

**Table 1: Overview of the $^{222}$Rn flux maps used in this study.**

| $^{222}$Rn flux map | Resolution | Soil moisture | Vertical resolution of soil parameters |
|---|---|---|---|
| GLDAS | 0.25° x 0.25°, daily | GLDAS-Noah (Rodell et al., 2004; Beauding and Rodell, 2020) | Soil moisture: 0-10-40-100 cm<br><br>Soil porosity: constant |
| ERA5 | 0.1° x 0.1°, daily | ERA5-Land (Muñoz-Sabater et al., 2021; Muñoz-Sabater, 2019) | Soil moisture: 0-10-40-100 cm<br><br>Soil porosity: constant |
| DWD/GLDAS | 1km x 1km, daily | DWD (Löpmeier, 1994; Herbst et al., 2021; Hammel and Kennel, 2001) | Soil moisture: 0-10-40-100 cm<br><br>Soil porosity: 0-10-40-100 cm |

**2.2 Atmospheric $^{222}$Rn concentration observations**

We use hourly $^{222}$Rn activity concentration observations from 17 European sites in 2021. The $^{222}$Rn measurements have been performed with several detector types, which are based on different measurement principles and assumptions. Figure 2 and Tab. 2 give an overview of the different observation sites and the detectors used. The main characteristics of the used detector types are compiled in Schmithüsen et al. (2017) and Grossi et al. (2020). For example, the radon detectors developed by the Australian Nuclear Science and Technology Organisation (ANSTO) are based on the so-called dual-flow-loop two-filter approach, which provides a direct measure of the $^{222}$Rn activity concentration (Griffiths et al., 2016). In this method, the sampled air passes through an initial filter that removes all ambient aerosols and $^{222}$Rn progenies. The filtered air is then

directed into a large delay volume (typically >1000 L) in which new $^{222}$Rn progenies ($^{218}$Po and $^{214}$Po) are formed. A second flow loop within the delay volume frequently circulates the sample of air through a second filter to ensure that all $^{222}$Rn progenies are collected on this filter (e.g., $^{218}$Po has a half-life of only 3 min) and their α-decay can be counted. Finally, the $^{222}$Rn activity concentration is calculated from the α-decay of the $^{222}$Rn progenies and the flow rate. Due to the large delay volume, ANSTO detectors have a relatively slow response time of about 45 min. Approximately 40 % of the signal is observed one hour after the radon pulse is delivered (Griffiths et al., 2016). This delay necessitates time response correction. To recover the true atmospheric signal, the ANSTO data from the sites in UK are corrected using a deconvolution procedure, following the methodology described in Griffiths et al. (2016) and further validated and standardized in Kikaj et al. (2025). This approach effectively recovers rapid variations in $^{222}$Rn activity concentrations, which are typically observed during periods of changing air mass origin, such as transitions between terrestrial and oceanic fetches or during the early morning and late evening hours when the planetary boundary layer (PBL) height changes rapidly. For the non-UK ANSTO data, we applied a first-order deconvolution by shifting the $^{222}$Rn data by one hour. Since only hourly averaged data from the afternoon period are used in this study, when atmospheric mixing is strongest and $^{222}$Rn concentrations are relatively stable, such a first-order deconvolution might be acceptable for our purposes. Furthermore, ANSTO detectors are affected by a gradually increasing background signal due to the accumulation of long-lived $^{210}$Po in the measurement chamber. This background is routinely monitored every 2–3 months and extracted from the signal using standardized correction procedures as part of the regular data processing protocol, ensuring the accuracy of reported $^{222}$Rn activity concentrations (Röttger et al., 2025; Kikaj et al., 2025).

In contrast, the Heidelberg Radon Monitor (HRM, Levin et al., 2002; Gachkivskyi and Levin, 2022) detectors installed at the German sites measure the $^{222}$Rn activity concentrations indirectly. In the atmosphere, the $^{222}$Rn progenies get attached to aerosols. The HRM detectors collect these atmospheric aerosols on a filter and measure the α-decay of the $^{222}$Rn progenies. In order to determine the $^{222}$Rn activity concentration from those measurements, assumptions must be made about the radioactive disequilibrium between $^{222}$Rn and its progenies (Schmithüsen et al., 2017). This disequilibrium depends on the height above ground (Jacobi and André, 1963). Inter-comparison studies between ANSTO and HRM detectors revealed that radioactive equilibrium between $^{222}$Rn and its daughter products is reached between ca. 50 - 100 m above ground (Schmithüsen et al., 2017; Grossi et al., 2020). We therefore applied equilibrium correction factors for observation sites with an air intake height below 90 m above ground. Furthermore, wet deposition of atmospheric aerosols as well as aerosol loss in long sampling lines can lead to artefacts in the HRM-based $^{222}$Rn activity concentrations (Xia et al., 2010; Grossi et al., 2016; Levin et al., 2017). To account for this, we (1) flagged the HRM data during situations with high air humidity >98% (>95% for the mountain sites HPB, SSL and TOH) as suggested by Gachkivskyi et al. (2025) and (2) applied for sites with sampling tubing lengths >15 m an aerosol loss correction as described in Levin et al. (2017). Finally, we averaged the half-hourly HRM $^{222}$Rn data to obtain hourly $^{222}$Rn observations. The corrected and moisture-selected HRM $^{222}$Rn observations are compiled in Fischer et al. (2024).

The radon detector installed at the Mace Head (MHD) observation site is another type of a one-filter radon monitor, again based on an indirect method of determining $^{222}$Rn activity concentration. It was developed at the Laboratoire des Sciences du Climat et de l'Environnement (LSCE) in France (Biraud, 2000) and uses a moving filter band system to collect and measure

the $^{222}$Rn progenies $^{218}$Po and $^{214}$Po. Since Schmithüsen et al. (2017) found a $^{214}$Po/$^{222}$Rn disequilibrium factor of 1.0 for this coastal site, we did not apply a disequilibrium correction to the MHD measurements. However, we again applied a humidity flag to account for the wet deposition of atmospheric aerosols. Note that the MHD $^{222}$Rn observations have a temporal resolution of only 2 hours.

Finally, we converted all $^{222}$Rn observations to the HRM detector scale in order to obtain a harmonized data set. Note that the ANSTO detector scale would result in approximately 11% larger $^{222}$Rn activity concentrations compared to the HRM detector scale (Schmithüsen et al., 2017). The LSCE measurements from MHD must be divided by 0.95 to convert them to the HRM scale (Schmithüsen et al., 2017). We assume an uncertainty of 0.5 Bq/m$^3$ for the hourly $^{222}$Rn observations used in the inversion to account for instrumental uncertainties, uncertainties associated with the required calibrations and corrections (Grossi et al.,

2020), as well as uncertainties in the background contributions from outside Europe, which we subtract from the $^{222}$Rn observations (see Sect. 2.3). As mentioned above, we only use observations from PBL sites during well-mixed situations in the afternoon (11-16 UTC), when the transport model is expected to show the best performance. In contrast, nighttime (23-04 UTC) observations are used for mountain sites when the impact of local (thermally induced) wind systems are expected to be negligible.


Note that there was a leak in the $^{222}$Rn inlet line at the Cabauw (CBW) site in 2021, which may have contaminated the $^{222}$Rn measurements at this site. However, as we found no obvious anomalies when comparing the observation-model differences at CBW with those at the nearby Lutjewad (LUT) site, we decided to use the CBW data in our study. Nevertheless, we investigated the impact of the CBW observations on the inversion results, which turned out to be small (see Appendix C).


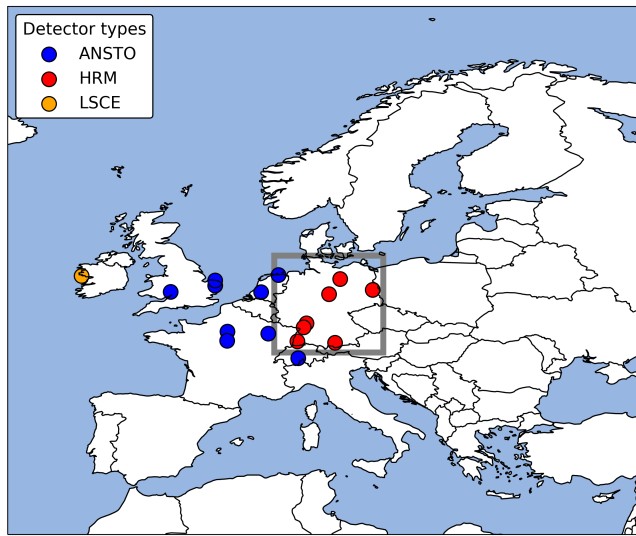

**Figure 2: Map of the European model domain. The Germany domain is indicated by the grey rectangle. The locations of the sites with [222]Rn observations used in the inversion are marked by coloured dots. The colours indicate the different radon detectors described in the text.**

**Table 2: Overview of the 17 European sites providing atmospheric [222]Rn measurements.**

| Site code | Site name | Coordinates (lat, lon, a.g.l.) | STILT release height (m a.g.l.) | FLEXPART/NAME release height (m a.g.l.) | Site class | Model uncertainty (Bq m[-3]) | Radon detector type | Number of days with observations (after 2σ-flagging) |
|---|---|---|---|---|---|---|---|---|
| CBW | Cabauw | 51.97°N, 4.93°E, 200m | 200 | 200 | T | 0.9 | ANSTO | 336 |
| GAT | Gartow | 53.07°N, 11.44°E, 132m | 132 (**) | 341 (**) | T | 0.9 | HRM | 323 |
| HEI | Heidelberg | 49.42°N, 8.67°E, 30m | 30 | 30 | U | 1.5 | HRM | 347 |
| HPB | Hohenpeißenberg | 47.80°N, 11.02°E, 93m | 300 (*) | 250 (*)/130 | T | 0.9 | HRM | 296 |

| JFJ | Jungfraujoch | 46.55°N, 7.99°E, 6m | 720 (*) | 530 (*)/1000 (*) | M | 0.45 | ANSTO | 322 |
|---|---|---|---|---|---|---|---|---|
| KIT | Karlsruhe | 49.09°N, 8.42°E, 200m | 200 | 200 | T | 0.9 | HRM | 341 |
| LIN | Lindenberg | 52.17°N, 14.12°E, 98m | 98 | 98 | T | 0.9 | HRM | 282 |
| LUT | Lutjewad | 53.40°N, 6.35°E, 60m | 60 | 60 | T | 0.9 | ANSTO | 338 |
| MHD | Mace Head | 53.33°N, 9.90°W, 24m | 24 | 10 | S | 0.45 | LSCE | 318 |
| OPE | Observatoire Pérenne de l'Environnement | 48.56°N, 5.50°E, 120m | 120 | 120 | T | 0.9 | ANSTO | 229 |
| RGL | Ridge Hill | 52.00°N, 2.54°W, 85m | 90 | 90 | T | 0.9 | ANSTO | 256 |
| SAC | Saclay | 48.72°N, 2.14°E, 100m | 100 | 100 | T | 0.9 | ANSTO | 330 |
| SSL | Schauinsland | 47.92°N, 7.92°E, 12m | 450 (*) | 250 (*)/10 | M | 0.45 | HRM | 212 |
| TAC | Tacolneston | 52.52°N, 1.14°E, 175m | 185 | 185 | T | 0.9 | ANSTO | 336 |
| TOH | Torfhaus | 51.81°N, 10.54°E, 110m | 110 (**) | 240 (*)/147 (**) | T | 0.9 | HRM | 249 |
| TRN | Trainou | 47.96°N, 2.11°E, 180m | 180 | 180 | T | 0.9 | ANSTO | 155 |

| WAO | Weybourne | 52.95°N, 1.12°E, 10m | 10 | 20 | | S | 0.45 | ANSTO | 335 |

T – tower sites; M – mountain sites; S – coastal sites; U – urban sites

(*) – high-altitude sites, for which a correction height was used in the model to account for the steep terrain

(**) – sites, for which the FLEXPART/NAME particle release height is >20 m different than that for STILT

## 2.3 Transport models

For our main inversion runs, we use the Stochastic Time-Inverted Lagrangian Transport model (STILT, Lin et al., 2003). In
Lagrangian transport models, an ensemble of numerical particles is released from each site every time step (here every hour of the measurements), and their back-trajectories are computed by confronting the particles with the underlying meteorological fields and a stochastic representation of turbulent motions. From the distribution of the back-trajectories, so-called footprints are deduced, which describe the sensitivity of the observation site to surface fluxes at the individual pixels in the catchment area of the site. The $^{222}$Rn activity concentrations $C(x_s, t_s)$ at the observation site $x_s$ and time $t_s$ can then be calculated by
convolving the $^{222}$Rn flux $F(x_j, t_i)$ of each grid cell $x_j$ ($j$=1…N, with N being the number of grid cells) and time step $t_i$ ($i$=1…T, with T being the maximal duration of the back-trajectories, i.e. 240 h in our case) with the footprint matrix $f$ and taking into account the radioactive decay of $^{222}$Rn with a lifetime of $\tau_{Rn}$= 5.52 d:

$$C(x_s, t_s) = \sum_j^N \sum_i^T f(x_s, t_s | x_j, t_i) \cdot F(x_j, t_i) \cdot \exp\left(-\frac{t_i}{\tau_{Rn}}\right) \tag{2}$$

STILT is driven with meteorological fields from the Integrated Forecasting System of the European Centre for Medium-Range Weather Forecasts (ECMWF-IFS), extracted at a spatial resolution of 0.25° x 0.25 ° (lat x lon) and a temporal resolution of 3 hours. Every hour, 100 particles are released from the observation sites and their back-trajectories are calculated for 10 days or until they leave the STILT model domain shown in Fig. 2. In STILT the horizontal resolution of the footprint matrix is dynamically reduced as the spatial distribution of the STILT particles increases. This prevents the under-sampling of surface
fluxes at times and regions where the STILT particles are distributed over extensive areas and have large gaps between each other (Gerbig et al., 2003). Therefore, this enables the usage of a small number of only 100 particles in STILT and thus reduces computational costs, while ensuring proper statistics. The final STILT footprints are mapped on a grid with a horizontal resolution of 0.25° x 0.25°.

In order to assess the influence of the transport model on the inversion results, we perform two additional inversion runs using footprints from two alternative Lagrangian transport models (see Tab. 3): the FLEXible PARTicle dispersion model (FLEXPART, Stohl et al., 2005; Pisso et al., 2019), and the Numerical Atmospheric-dispersion Modelling Environment (NAME, Jones et al., 2007). Similar to STILT, FLEXPART is also driven by meteorological analysis and forecasts from the operational ECMWF-IFS high-resolution (HRES) runs, available hourly at 0.1° x 0.1° resolution for the European domain and

3-hourly at 0.5° x 0.5° for the rest of the globe. With FLEXPART, 20'000 particles are released every hour and the back-trajectories are calculated for 30 days or when leaving a domain encompassing parts of North America, the North Atlantic and Europe. The resulting footprints are mapped on a grid with a horizontal resolution of 0.234° x 0.352° (lat x lon). NAME utilises meteorological data from the UK Met Office Unified Model (UM; Cullen, 1993). As with FLEXPART, 20'000 particles are released every hour and the back-trajectories are calculated for 30 days. The NAME footprints also have a horizontal resolution

of 0.234° x 0.352°. The FLEXPART and NAME footprints have initially been calculated for another project and another model domain. Therefore, the back-trajectories were calculated for 30 days, to ensure that most of the particles have left this larger model domain. However, to resolve $^{222}$Rn fluxes in Europe, a 10-day simulation, as is done with STILT, is sufficient.

Moreover, the FLEXPART and NAME footprints that we are using as alternative footprints for this study are already post-

processed, meaning that they are (1) corrected for the $^{222}$Rn decay and (2) aggregated over the duration of the back-trajectories (i.e., over 30 days or until the particles have left the model domain). This means that we have no temporal information about the particle distributions any more. However, since the $^{222}$Rn flux maps have a daily resolution, we need to know for which $^{222}$Rn flux time period the time-aggregated footprints are mainly sensitive. To investigate this, we make use of the time-resolved STILT footprints and perform two types of $^{222}$Rn simulations for three different sites (continental, coastal, and mountain sites;

see Fig. A1). In the first simulation, we modelled $^{222}$Rn concentrations by mapping the time-resolved STILT footprints with the daily $^{222}$Rn fluxes (as described in Eq. 2). For the second type of simulation, we first aggregated the STILT footprints over the duration of the back-trajectories (i.e., over 10 days) and corrected them for the $^{222}$Rn decay to mimic the post-processed FLEXPART and NAME footprint products. We then mapped these time-aggregated STILT footprints with $^{222}$Rn fluxes, which were averaged over different time intervals ranging from 1 to 10 days (see Fig. A1). By comparing both types of simulations

we can investigate for which $^{222}$Rn flux time interval the time-aggregated footprints are most sensitive. We found that the $^{222}$Rn concentration differences between the first and the second simulations show a low bias and a small standard deviation when the time-aggregated footprints are mapped with $^{222}$Rn fluxes that are averaged over 3 days, indicating that the time-aggregated footprints are mainly representative for the average $^{222}$Rn fluxes of the first 3 days before particle release. Therefore, we average the $^{222}$Rn fluxes over 3 days when mapping them with the FLEXPART and NAME footprints. However, we want to

emphasise that we use the time-resolved STILT footprints to perform our main analyses in this study and that we apply the time-aggregated FLEXPART and NAME footprints only to investigate the impact of different transport models on the inversion results, by which means we assess the robustness of our results.

**Table 3: Overview of the different transport models**

| Model | Meteorology | Number of particles | Length of back-trajectories | Horizontal resolution of footprint (lat x lon) | Temporal resolution of footprint |
|---|---|---|---|---|---|

| STILT | ECMWF, 3-hourly, 0.25° x 0.25° | 100 | 10 days | Highest resolution: 0.25° x 0.25°; dynamic coarsening of footprint resolution (see text) | hourly |
|---|---|---|---|---|---|
| FLEXPART | ECMWF, hourly, 0.1° x 0.1° | 20'000 | 30 days | 0.234° x 0.352° | time-aggregated |
| NAME | UK Met Office Unified Model | 20'000 | 30 days | 0.234° x 0.352° | time-aggregated |


The back-trajectories of the numerical particles are only calculated whilst the particles remain within the European model domain. For particles that leave the model domain, lateral boundary conditions are needed. For this purpose, we have constructed a simple global $^{222}$Rn flux map, assuming a constant $^{222}$Rn flux of 1 atom cm$^{-2}$ s$^{-1}$ (21 mBq m$^{-2}$ s$^{-1}$) over land

surfaces south of 60°N, a halved $^{222}$Rn flux of 0.5 atoms cm$^{-2}$ s$^{-1}$ (10.5 mBq m$^{-2}$ s$^{-1}$) over land in the higher latitudes >60°N, and a vanishing $^{222}$Rn flux in permafrost regions. The much smaller $^{222}$Rn fluxes from the ocean were neglected. We use the Eulerian global atmospheric Tracer Model (TM3, Heimann and Körner, 2003) and this global $^{222}$Rn flux map to simulate hourly $^{222}$Rn activity concentrations for each European grid cell. A $^{222}$Rn background concentration is then determined for each site and hour by averaging the modeled $^{222}$Rn activity concentrations in the respective grid cells of the endpoints of the STILT

back-trajectories after taking radioactive decay into account. Due to the relatively short $^{222}$Rn atmospheric lifetime, these background $^{222}$Rn activity concentrations are typically quite low (the median of the mean background contribution across all sites is <15%). We subtract this modelled $^{222}$Rn background from the $^{222}$Rn observations and use the resulting $^{222}$Rn excess concentrations to constrain the $^{222}$Rn fluxes in Europe. Note that we use the same $^{222}$Rn excess concentrations when we apply the time-aggregated FLEXPART and NAME footprints in the inversion.


To investigate the impact of the simulated background on the $^{222}$Rn flux results, we perform two additional inversion runs using alternative $^{222}$Rn backgrounds. For the first run, we replace the TM3-simulated $^{222}$Rn concentration field by the $^{222}$Rn concentration field provided by the Copernicus Atmosphere Monitoring Service (CAMS) global reanalysis (EAC4; Inness et al., 2019) and again use the endpoints of the STILT trajectories to calculate an alternative (decay-corrected) $^{222}$Rn background.

For the second run, we fully neglect the Rn background, i.e. we assume that it is zero.

### 2.4 Inversion setup

In this study, we use the CarboScope-Regional (CSR) inversion system described in Rödenbeck et al. (2003, 2009) to constrain the process-based $^{222}$Rn fluxes with the atmospheric $^{222}$Rn observations. In the following, we provide a brief overview of the

CSR inversion system, with a specific focus on the aspects that are particularly relevant to the $^{222}$Rn inversion. For more technical details about the inversion algorithm and how the iterative solution is found, we refer the reader to Rödenbeck (2005).

In the CSR system, the Bayesian cost function is implemented using a "linear statistical flux model" (Rödenbeck et al., 2009):

$$x = x_{fix} + \mathbf{F}p \tag{3}$$

The flux field $x$ is written as the sum of a fixed flux component $x_{fix}$, which is the mean of the prior flux field $x_{prior}$, and the product between a matrix $\mathbf{F}$ and a vector $p$ with adjustable parameters. By construction, the a priori parameters in $p$ have zero mean, unit variance, and are uncorrelated. The columns of $\mathbf{F}$ represent spatio-temporal flux patterns, which are scaled by the elements of $p$. The prior error covariance matrix $\mathbf{B}$ is then given by $\mathbf{B}=\mathbf{FF}^T$, and the cost function J($p$) is:

$$J(p) = \tfrac{1}{2}p^T p + \tfrac{1}{2}(c_{obs} - \mathbf{H}x)^T \mathbf{R}^{-1}(c_{obs} - \mathbf{H}x) \tag{4}$$

The second term in Eq. 4 shows the data constraint, and the first term describes the prior constraint.

The quadratic Bayesian cost function is minimized by applying a conjugate gradient algorithm that allows large state vectors. The cost function includes a model-data mismatch vector that contains the hourly differences between the observed ($c_{obs}$) and modelled ($c_{mod}=\mathbf{H}x$, with $\mathbf{H}$ being the transport matrix) $^{222}$Rn activity concentrations from all 17 sites. The model-data mismatch vector is weighted with a covariance matrix $\mathbf{R}$ containing the uncertainties of the $^{222}$Rn observations and the transport model. As already mentioned, we assume an uncertainty of 0.5 Bq m$^{-3}$ for the hourly $^{222}$Rn observations. The uncertainty of the transport model is chosen dependent on the type of observation site (Rödenbeck 2005). For example, continental tower sites can typically be better represented in the model than sites such as Heidelberg, which is located in a narrow river valley with complex local circulation. Depending on the site, we therefore assume a transport model uncertainty of ca. 0.5 Bq m$^{-3}$ to 1.5 Bq m$^{-3}$ (see Tab. 2). The total model-data mismatch uncertainty is obtained by adding the observation and transport model uncertainties quadratically. To account for temporal correlations between model-data mismatch within one week, we applied the so-called data density weighting proposed by Rödenbeck (2005). This inflates the uncertainty of the model-data mismatch by the square root of the number of observations within a week. One week is the typical timescale for synoptic events, in which the $^{222}$Rn observations are expected to be temporally correlated. As mentioned above, we only use afternoon observations (or nighttime observations for mountain sites), when the atmosphere is typically well mixed. In addition, we applied a 2$\sigma$-filtering to these data (as described in Rödenbeck et al., 2018) to exclude the observations with the largest model-data mismatch, as these are considered to be inadequately represented by the transport model.

The Bayesian approach adds a priori information to stabilize the solution. We use the process-based $^{222}$Rn flux maps described in Sect. 2.1, as well as a map with spatially and temporally constant fluxes over the European continent ("flat prior") as a priori estimates. Due to the large differences between the $^{222}$Rn flux maps, we assume an a priori uncertainty of 100% for the European $^{222}$Rn fluxes and for a time scale similar to the temporal correlation length. In the standard setting of our inversion

system, we assume that the a priori flux errors are spatially correlated over a length scale of about 400 km, and we choose a temporal correlation length of 3.5 days ("Filt52T" in CarboScope notation). In Sect. 3.2.2, we investigate how changes to these settings affect the inversion results. The a priori uncertainties are described by the a priori covariance matrix **B**, which
determines the a priori constraint. The ratio between a priori and data constraint determines how strongly the solution is regularized by the a priori information. The inversion system minimizes the model-data mismatch by scaling the $^{222}$Rn fluxes, taking into account the model-data mismatch and the a priori uncertainties. Overall, we use the CSR system to determine daily $^{222}$Rn fluxes for the whole year 2021 at a spatial resolution of 0.25°.

## 2.5 Overview of the different inversion runs

To investigate how robust the $^{222}$Rn flux estimates are, we perform several inversion runs using different prior flux estimates, varied parameter settings in the inversion setup, and different transport models (see an overview in Tab. 4).

**Table 4: Overview of the different inversion runs.**

| Sensitivity runs | Transport model | Prior fluxes | Prior uncertainty | Temporal correlation length | Spatial correlation length | $^{222}$Rn observation scale | $^{222}$Rn background |
|---|---|---|---|---|---|---|---|
| flat | STILT | flat prior | 100% | 3.5 days | ca. 400 km | HRM | TM3 |
| GLDAS | STILT | GLDAS | 100% | 3.5 days | ca. 400 km | HRM | TM3 |
| ERA5 | STILT | ERA5 | 100% | 3.5 days | ca. 400 km | HRM | TM3 |
| DWD | STILT | DWD/GLDAS | 100% | 3.5 days | ca. 400 km | HRM | TM3 |
| flat_50 | STILT | flat prior | 50% | 3.5 days | ca. 400 km | HRM | TM3 |
| flat_7days | STILT | flat prior | 100% | 7 days | ca. 400 km | HRM | TM3 |
| flat_200km | STILT | flat prior | 100% | 3.5 days | ca. 200 km | HRM | TM3 |
| flat_ANSTO | STILT | flat prior | 100% | 3.5 days | ca. 400 km | ANSTO | TM3 |
| flat_CAMS | STILT | flat prior | 100% | 3.5 days | ca. 400 km | HRM | CAMS-EAC4 |
| CSR-FLEX | FLEXPART | flat prior | 100% | 3.5 days | ca. 400 km | HRM | TM3 |
| CSR-NAME | NAME | flat prior | 100% | 3.5 days | ca. 400 km | HRM | TM3 |

As soil moisture controls temporal changes of the diffusion coefficient within the soil and is therefore expected to be the main driver of the temporal variability of the $^{222}$Rn flux, we want to investigate what information about soil moisture is contained in the $^{222}$Rn observations. For this purpose, we constructed a "flat" prior with spatially and temporally constant $^{222}$Rn fluxes over land. Such a flat-prior inversion does not use any a priori information on soil moisture variability. This means that the a posteriori flux variability is only caused by the signals in the $^{222}$Rn observations (possibly with spurious contributions from

variations in the transport model error). To investigate the extent to which variations in the $^{222}$Rn flux are caused by changes in soil moisture, we will calculate the temporal correlation between the daily a posteriori flux of a domain covering Germany and the daily GLDAS-Noah and ERA5-Land, respectively, soil moisture average of the same domain (see Sect. 3.3).

## 3 Results

### 3.1 Comparison of the process-based $^{222}$Rn flux maps

Figure 3 shows a comparison between the three process-based $^{222}$Rn flux maps. The GLDAS and ERA5 $^{222}$Rn flux maps show similar $^{222}$Rn hotspot regions, e.g. on the Iberian Peninsula and in Italy, which can be explained by the high uranium activity concentrations there (see Fig. 3a,b). Note that these $^{222}$Rn flux maps differ only in the soil moisture and porosity data, but not in the underlying uranium map. Compared to the ERA5-Land soil moisture, the GLDAS-Noah soil moisture leads to larger annual mean $^{222}$Rn fluxes in central Europe but to smaller $^{222}$Rn fluxes in Scandinavia (Fig. 3c). Overall, the annual mean flux

differences between both maps can be as large as the fluxes themselves. The DWD/GLDAS $^{222}$Rn flux map has a higher spatial resolution for Germany (Fig. 3e). The DWD soil moisture and porosity leads to lower annual mean $^{222}$Rn fluxes in the southern part of Germany and to slightly higher $^{222}$Rn fluxes in northern Germany than the $^{222}$Rn fluxes based on GLDAS-Noah soil moisture and porosity data (Fig. 3f).

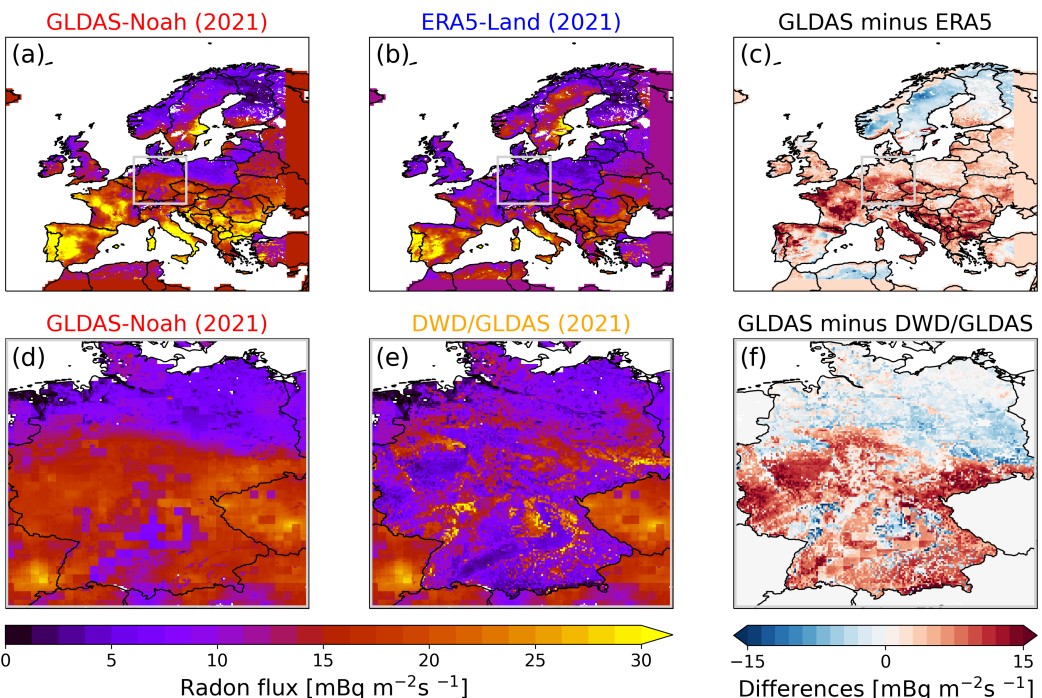

**Figure 3: Annual mean $^{222}$Rn fluxes in 2021 for the European model domain (a-b) and for the Germany domain (d-e) based on GLDAS-Noah (a, d), ERA5-Land (b), and DWD/GLDAS (e) soil moisture data. The DWD/GLDAS $^{222}$Rn flux map (e) is based on DWD soil moisture and porosity data for Germany and GLDAS-Noah soil moisture and porosity data for outside Germany. Panels**

(c) and (f) show the annual mean differences between the GLDAS-Noah and ERA5-Land based [222]Rn fluxes and the GLDAS-Noah and DWD/GLDAS based [222]Rn fluxes, respectively.


## 3.2 Top-down evaluation of the GLDAS and ERA5 [222]Rn flux maps

### 3.2.1 Comparison between a posteriori and process-based [222]Rn fluxes

We start by evaluating the existing [222]Rn flux maps from Karstens and Levin (2022a,b), which are based on the GLDAS-Noah and ERA5-Land soil moisture data. On an annual average, the inversion increases the process-based a priori [222]Rn fluxes in

central Europe and decreases them over the British Isles (see Fig. 4a,b). This corresponds to the negative model-data mismatches caused by the a priori fluxes at most of the observation sites in central Europe and the positive model-data mismatches at the Irish site Mace Head (MHD) and the British site Weybourne (WAO; see Fig. B1). The negative flux adjustments over the British Isles occur throughout the whole year (see Fig D1). They could be due to boundary effects, as the British Isles are most affected by potential biases in the [222]Rn background due to the prevailing westerly winds in Europe. The

ERA5 a priori fluxes lead to more negative model-data mismatches at most of the continental sites than the GLDAS a priori fluxes (Fig. B1). Consequently, the ERA5 fluxes in central Europe are increased much more by the inversion than the GLDAS [222]Rn fluxes, bringing the posterior fluxes closer to each other than the priors (Fig. 4c). While in the case of the GLDAS inversion there are also some negative flux adjustments in France during summer, the ERA5 inversion shows positive flux adjustments in France and Germany throughout the whole year (Fig. D1).


The posterior flux estimate of the flat-prior inversion shows some substantial differences of up to roughly 50% for individual months compared to the posterior estimates based on the GLDAS and ERA5 prior fluxes (compare grey curve with red and blue curve in Fig. 4c). In particular, the European flux estimates only show a substantial seasonal cycle when the process-based prior fluxes are used, which in turn also exhibit a pronounced seasonal cycle. In contrast, using the flat prior results in

much weaker seasonal variability in the posterior flux. This demonstrates that the continental-scale [222]Rn flux estimates and their seasonal variability are strongly influenced by the prior information. This can be explained by the sparse [222]Rn data coverage in Europe, which is insufficient to reliably constrain the seasonal variability of the European [222]Rn fluxes. In the following we will therefore focus our analysis on a central European domain around Germany, which is covered well by the [222]Rn observations.


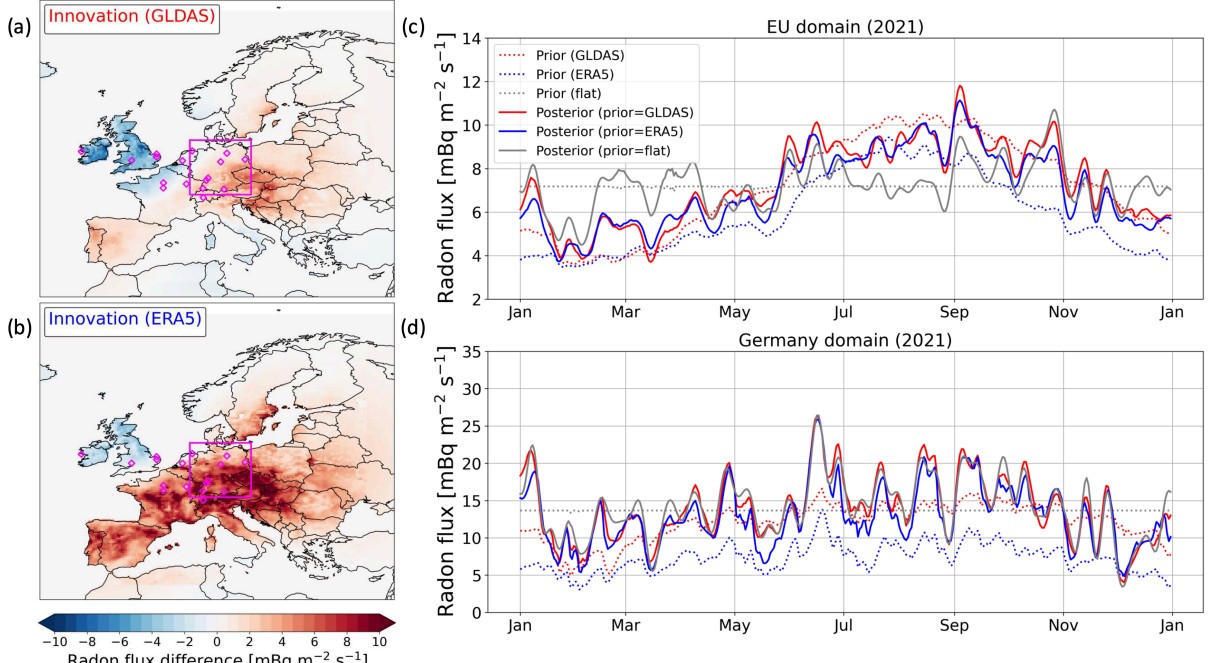

**Figure 4: Results of the CSR-STILT inversion runs with GLDAS (red), ERA5 (blue), and flat (grey) a priori $^{222}$Rn fluxes. Panels (a) and (b) show the annual mean innovation (a posteriori minus a priori $^{222}$Rn flux differences) for the inversion runs based on the prior fluxes from GLDAS and ERA5, respectively. Panels (c) and (d) show the time series for the full year of the a priori (dotted line) and the a posteriori (solid line) $^{222}$Rn fluxes averaged over the entire European domain (c) and over the Germany domain (d), respectively, for 2021. The Germany domain is depicted by the magenta rectangle in the maps in panels (a) and (b).**

In this Germany domain, the flat-prior inversion yields very similar flux estimates than the inversions based on the GLDAS and ERA5 a priori fluxes (Fig. 4d). The annual means of the three different posterior fluxes agree within 10%, which is rather small considering the two process-based a priori fluxes in the Germany domain differ by more than 60% on average in 2021. Also, the temporal variability of the posterior fluxes is comparable; the normalized standard deviations of the three different posterior fluxes agree within 10%. Furthermore, the different inversion runs produce similar seasonal variations, which are comparable with the seasonal variation in the process-based prior fluxes. While the soil moisture driven GLDAS and ERA5 prior fluxes are 29% and 37% higher in the summer half-year of 2021 than in the winter half-year, the corresponding posterior fluxes show 36% and 27%, respectively, higher fluxes in summer than in winter. The flat-prior inversion leads to a very similar result: the flux estimates in the summer half-year are 30% higher than in the winter half-year. Overall, this illustrates that the flux estimates in the Germany domain are indeed well constrained by the observations and less affected by the prior information. Therefore, in the following, we will use the flat-prior inversion results, which are not based on additional information about temporal and spatial flux variability (and are thus based on the least a priori information), to evaluate the performance of the process-based $^{222}$Rn flux maps.

In the Germany domain, the annual mean a posteriori $^{222}$Rn flux (of the flat-prior inversion) is about 20% and almost 100% larger than the GLDAS and ERA5 process-based $^{222}$Rn fluxes, respectively, indicating that, in particular, the ERA5 $^{222}$Rn fluxes might be too low in central Europe. In addition, the a posteriori flux shows a higher temporal variability than the bottom-up fluxes. The normalized standard deviation of the daily a posteriori $^{222}$Rn flux is about 10% (ERA5) and 35% (GLDAS) higher than the normalized standard deviation of the process-based fluxes in the Germany domain. In Sect. 3.4, we investigate if we can increase the process-based $^{222}$Rn fluxes and their temporal variability by using higher resolution soil moisture and porosity data for Germany. Overall, the inversion reduces the annual mean and the standard deviation of the model-data mismatch at almost all sites (Fig. B1).

### 3.2.2 Sensitivity of inversion results to model configuration

After having shown that the different a priori fluxes (GLDAS, ERA5, flat) lead to only small changes in the a posteriori fluxes in the Germany domain, we want to investigate how robust these inversion results are. For this, we conduct several inversion runs using different settings, to assess sensitivity (see Tab. 4).

The various inversion runs performed with CSR-STILT lead to very similar a posteriori $^{222}$Rn fluxes in Germany (see Fig. 5). These runs differ in terms of the prior information used, the inversion's uncertainty parameters, the scale of the $^{222}$Rn observations, and the simulated $^{222}$Rn background. Most of these sensitivity tests yield a posteriori fluxes with an annual mean difference well below 10% compared to the a posteriori flux of the flat-prior inversion run based on the standard setting (described in the first row of Tab. 4). This shows that we indeed get robust results for the Germany domain, which is well covered by $^{222}$Rn observation sites. There is one exception to this result however, inversion run 'flat_ANSTO', which yields an a posteriori flux that has an almost constant offset of 13% compared to the respective inversion run with the standard setting (magenta curve in Fig. 5). For this run we have applied a different $^{222}$Rn observation scale (ANSTO scale instead of HRM scale, see Sect. 2.2; note that the simulated background $^{222}$Rn concentration is the same as for the inversions with the HRM scale). Since this ANSTO scale leads to 11% higher $^{222}$Rn activity concentrations than the HRM scale, such an offset in the $^{222}$Rn fluxes is to be expected. This illustrates the urgent need for well-calibrated and SI-traceable $^{222}$Rn observation data sets.

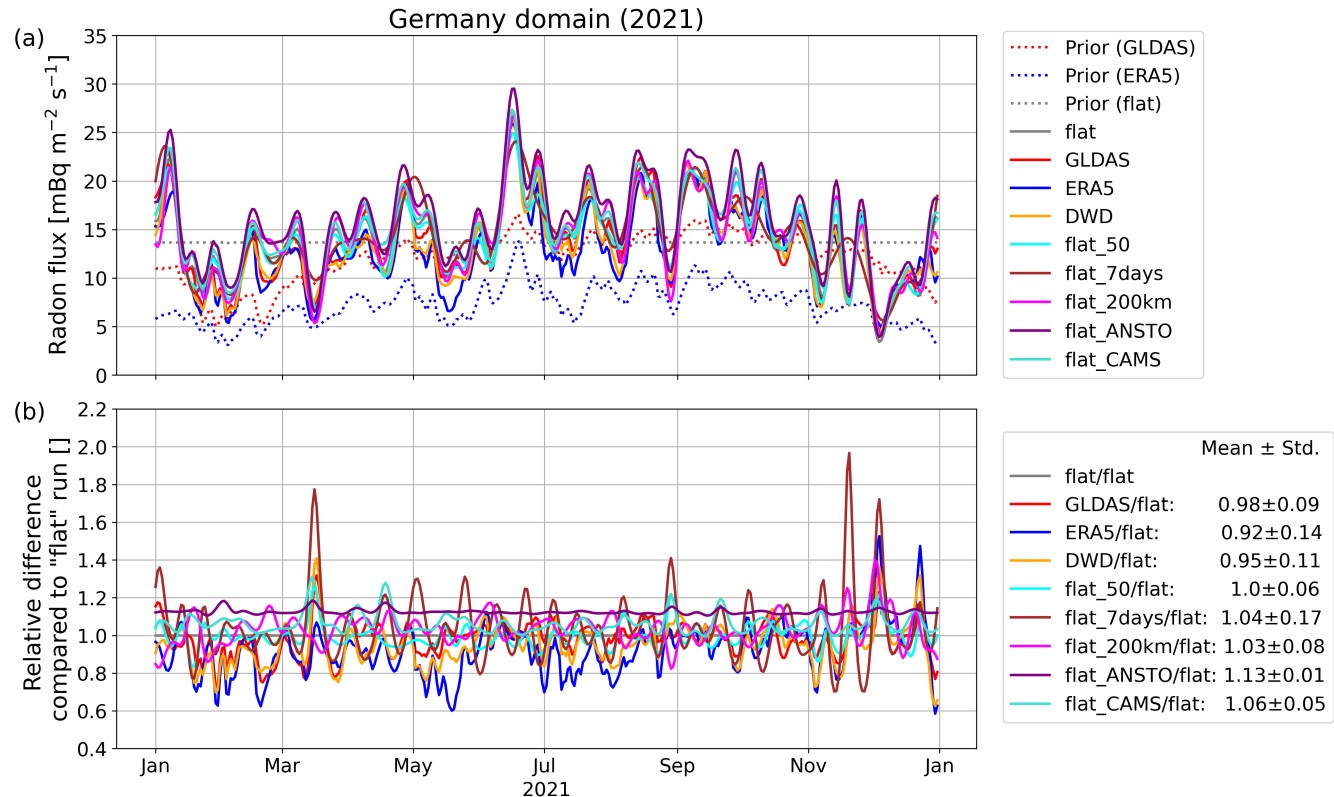

**Figure 5: (a) Daily a posteriori ²²²Rn fluxes in the Germany domain for different inversion runs performed with CSR-STILT. Panel (b) shows the relative difference between the respective a posteriori fluxes and the flat-prior inversion run (grey curve in (a)) with the standard parameter settings described in Sect. 2.4. The different inversion runs are described in Tab. 4.**

Next, we analyse the results of the CSR-FLEX and CSR-NAME inversions, for which FLEXPART and NAME footprints are used instead of the STILT footprints. When the GLDAS prior ²²²Rn fluxes are mapped with the NAME footprints, the resulting ²²²Rn concentrations strongly overestimate the observations from the mountain site Schauinsland (SSL) by ca. 2.7 Bq m⁻³ on an annual average (see Fig. B1). Such large overestimations are not observed at other sites and in the STILT and FLEXPART simulations. This could be due to the fact that the NAME model, unlike STILT and FLEXPART, did not use an elevated release height for the SSL site (see Tab. 2) and therefore has difficulty in correctly representing the steep terrain there. Therefore, we excluded the observations from SSL in the case of the CSR-NAME inversion to avoid unrealistic flux adjustments in southwest Germany.

For most of the continental sites, FLEXPART and NAME show higher surface influences, and thus larger prior $^{222}$Rn concentrations and smaller model-data mismatches than STILT (Fig. B1). Averaged over all sites in the Germany domain, the FLEXPART simulations lead to 46% and 18% (for GLDAS and ERA5 $^{222}$Rn flux, respectively) and the NAME simulations

lead to 47% and 32% smaller model-data mismatches than STILT (average of NAME model-data mismatch without SSL). Consequently, the CSR-FLEX and CSR-NAME inversions lead to smaller $^{222}$Rn flux adjustments in central Europe than STILT. On an annual average, the posterior $^{222}$Rn flux based on the flat prior is 5% and 12% smaller for the CSR-FLEX and CSR-NAME inversions, respectively, than for the CSR-STILT inversions in the Germany domain (see Fig. 6). However, there seems to be a slight seasonal cycle in the difference. In the winter half-year the FLEXPART and NAME based a posteriori

$^{222}$Rn fluxes are within 3% of the STILT based fluxes, whereas during summer the FLEXPART and NAME based a posteriori $^{222}$Rn fluxes are 12% and 23% lower than the STILT based flux, respectively. This seasonal cycle in the difference could indicate seasonal variations in the model performances, which should be investigated further. As a consequence, the FLEXPART and NAME runs exhibit a smaller seasonal cycle in the $^{222}$Rn flux than the STILT run.

Overall, these results show that changing the transport model leads to some deviations in the a posteriori fluxes, but these are comparable to the deviations caused by varying the inversion parameter settings, as shown in Fig. 5. Moreover, these transport model-based deviations are not larger than the annual flux differences caused by the choice of the $^{222}$Rn observation scale.

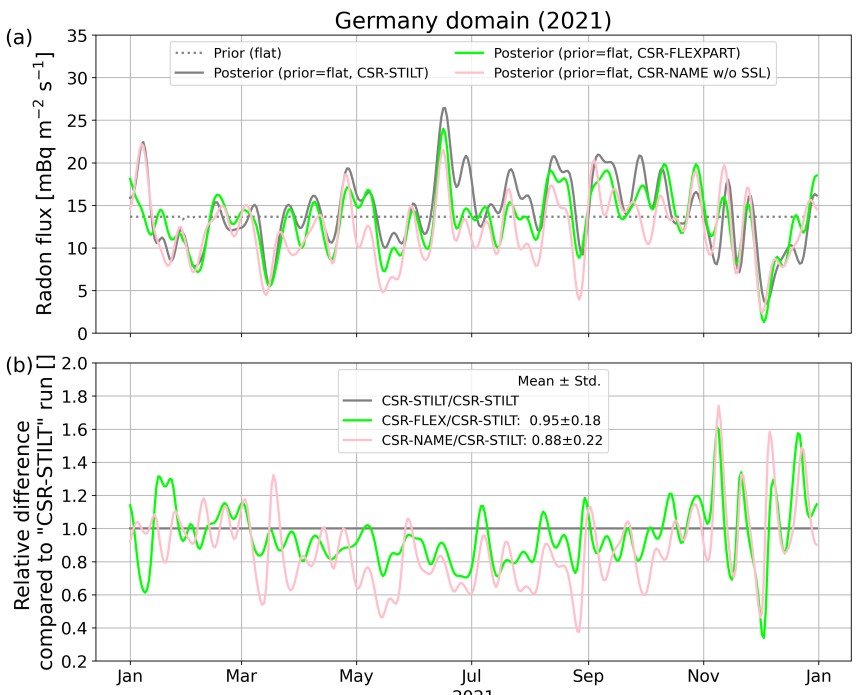

**Figure 6: Comparison between the CSR-STILT (grey), CSR-FLEX (green), and CSR-NAME (pink) inversion results for the**
**Germany domain. All inversions were performed with the flat a priori flux (grey dotted line in (a)). In the case of the CSR-NAME inversion, the observations from SSL weren't used ("w/o SSL") due to unexpectedly large prior model-data mismatches at that site**

**(see text). Panel (a) shows the daily $^{222}$Rn fluxes and panel (b) shows the relative difference between the different a posteriori fluxes and the flat-prior inversion run performed with CSR-STILT (solid grey curve in (a)).**

## 3.3 What soil moisture information is in the $^{222}$Rn observations?

Figure 7 shows the temporal correlation between the daily a posteriori $^{222}$Rn flux and soil moisture data, both averaged over the Germany domain. For both soil moisture products, we obtain quite strong anti-correlations between about -0.6 for GLDAS-Noah and even -0.8 for ERA5-Land. This anti-correlation means that we get high $^{222}$Rn fluxes when soil moisture is low, which makes sense as soil pores are less filled with water in dry conditions and the diffusion coefficient of $^{222}$Rn is higher in air than in water. In addition, the stronger anti-correlation in the case of ERA5-Land might indicate that this reanalysis product describes the temporal variability of the soil moisture better than GLDAS-Noah. We interpret the existence of meaningful correlations also as a confirmation that the inversion is indeed picking up real flux variations (even though spurious correlations due to the relationships of soil moisture and atmospheric mixing cannot be excluded either).

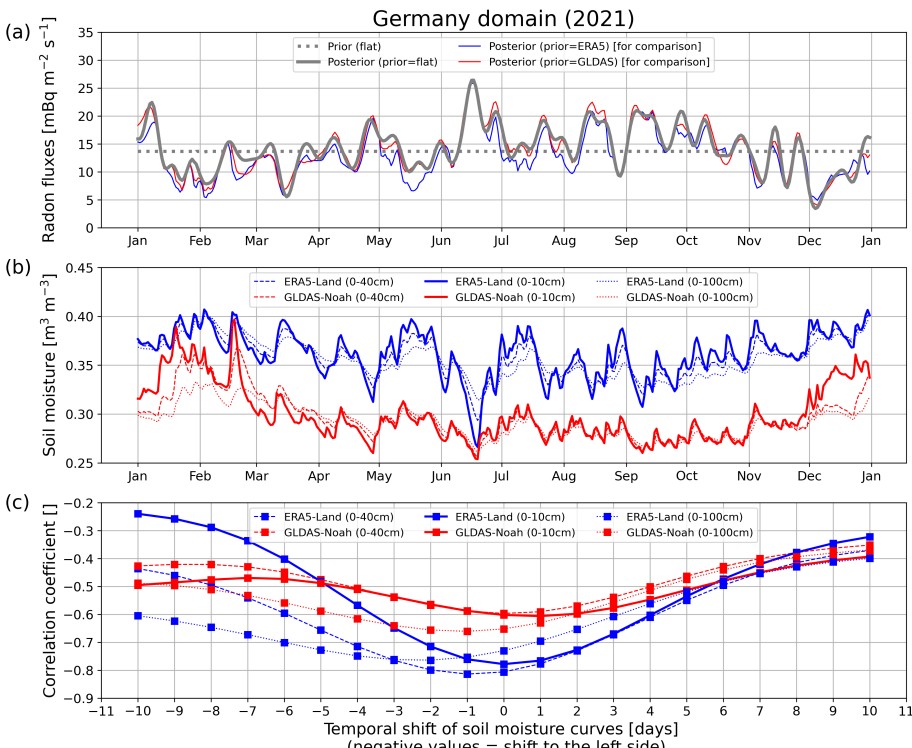

**Figure 7: (a) Flat prior (in grey, dotted) and associated a posteriori flux (in grey, solid) for the Germany domain, together with the ERA5 and GLDAS a posteriori fluxes for comparison. (b) ERA5-Land (blue) and GLDAS-Noah (red) soil moisture within the Germany domain for the upper 0-10 cm (solid), 0-40 cm (dashed), and 0-100 cm (dotted) of the soil column. (c) Correlation between the posterior flux curve of the flat-prior inversion shown in (a) and the different soil moisture curves shown in (b), calculated for different time lags of the soil moisture curves.**

## 3.4 Evaluating soil depth sensitivity in $^{222}$Rn flux

An open question is why the process-based flux maps driven by soil moisture fields show less variations than the flux estimates from the inversion. Karstens et al. (2015) estimated that the soil parameters of the top 100 cm of soil are most important for $^{222}$Rn flux at the surface, and Karstens and Levin (2024) used the average soil moisture of the top 40 cm of soil in their $^{222}$Rn flux model. To investigate systematically over which soil depth interval the soil moisture data should be averaged, we calculated the average soil moisture of the top 100 cm, top 40 cm, and top 10 cm of soil for both reanalysis products GLDAS-Noah and ERA5-Land, and calculated time-lagged correlations with the a posteriori flux of the flat-prior inversion (Fig. 7c). The average soil moisture of the top 10 cm shows the largest temporal variability, and increases rapidly when it rains. The increase of the average soil moisture of the top 40 cm and top 100 cm is slightly delayed after a rain event (cf. Fig7 and Fig. G1).

If we average the soil moisture data of the top 40 cm of soil, as done in Karstens and Levin (2024), the maximum anti-correlation between the soil moisture time series and the posterior $^{222}$Rn flux of the flat-prior inversion is reached when the soil moisture curve is shifted by one day, which means that the soil moisture lags the $^{222}$Rn flux by one day. This time lag even increases to 2-3 days if the soil moisture of the top 100 cm of the soil is averaged. So, the $^{222}$Rn flux responds faster to rain or drought events than the average soil moisture of the top 40 or 100 cm of soil, suggesting that the average soil moisture responds in a delayed way. However, if only the soil moisture of the top 10 cm is averaged, the time lag between soil moisture and $^{222}$Rn flux disappears. This could indicate that the variability of the $^{222}$Rn flux is mainly caused by the soil moisture variability in the top 10 cm of the soil. This leads to the question: Does the $^{222}$Rn flux model produce larger temporal variations if driven by average soil moisture data from the top 10 cm of soil only?

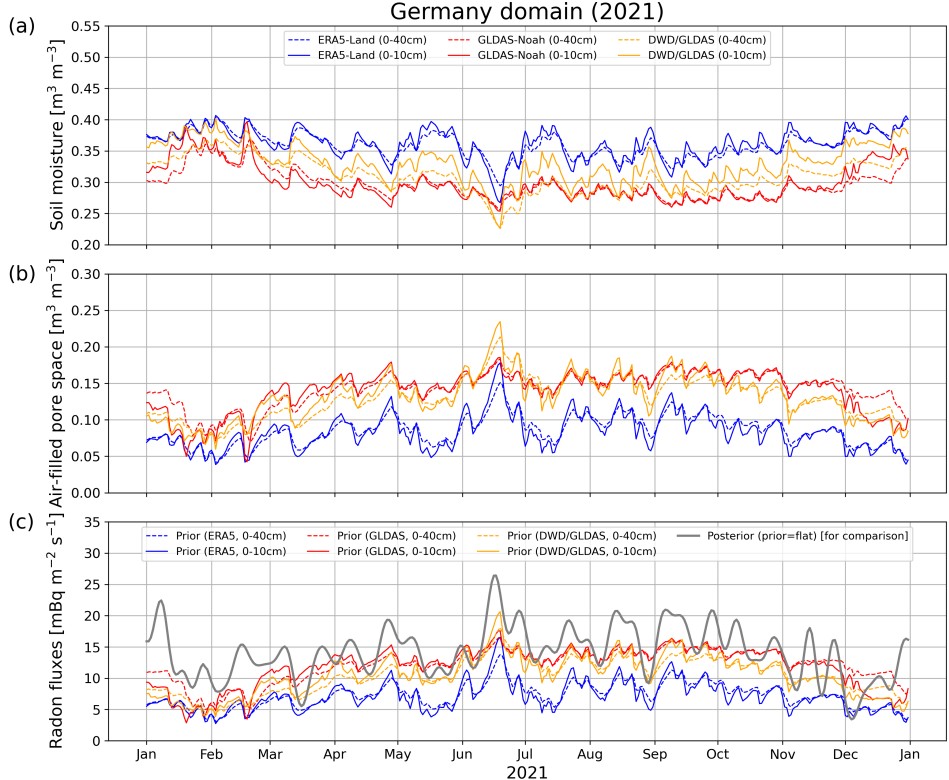

**Figure 8: Soil moisture (a), air-filled pore space (b) and process-based $^{222}$Rn fluxes (c) based on the ERA5-Land (blue), GLDAS-Noah (red), and DWD/GLDAS (orange) soil moisture and porosity data averaged over the top 10 cm (solid) and the top 40 cm (dashed) of the soil column. The a posteriori flux (grey) based on the flat prior is shown in (c) for comparison.**

Indeed, the average soil moisture in the top 10 cm of the soil shows a higher temporal variability than the average soil moisture in the top 40 cm, leading to an increased temporal variability of the $^{222}$Rn flux (see Fig. 8). Since the average soil moisture of the top 10 cm is higher during rain events and lower during dry periods than the average soil moisture of the top 40 cm, the $^{222}$Rn fluxes based on the 10 cm average soil moisture tend to be higher during dry periods and lower during rain events compared to the $^{222}$Rn fluxes based on the 40 cm average soil moisture (see Fig. G1). The daily $^{222}$Rn fluxes can differ by up to roughly 30% (ERA5) and 60% (GLDAS) if the 10 cm soil moisture is used instead of the 40 cm soil moisture. The normalized standard deviation of the $^{222}$Rn fluxes based on the top 10 cm soil moisture data is 22% (ERA5) and 24% (GLDAS) higher compared to the respective fluxes based on the top 40 cm soil moisture data. Consequently, the bottom-up $^{222}$Rn fluxes based on the top 10 cm soil moisture data show a similar temporal variability as the flux estimate of the flat-prior inversion (the normalized standard deviations of the bottom-up fluxes and the inversion result agree within about 10%). Nevertheless, using the top 10 cm soil moisture data instead of 40 cm has, for both reanalysis products, only a very minor impact on the annual mean $^{222}$Rn fluxes (in fact the soil moisture data from the top 10 cm of soil lead to ca. 2-3% lower annual mean $^{222}$Rn fluxes). It should be noted that both reanalysis products use the same porosity values for all soil layers, so they assume that porosity does not change with soil depth.

Next, we evaluated if we could further improve the process-based $^{222}$Rn fluxes by using a high-resolution soil moisture data product from the DWD model for Germany, which is based on vertically resolved porosity data (see Sect. 2.1). The average DWD soil moisture in the top 10 cm lies between the ERA5-Land and GLDAS-Noah soil moisture and is on average even 0.02 m$^3$/m$^3$ higher than in the top 40 cm. However, the porosity in the top 10 cm is also higher than in the top 40 cm, which could be explained by a layer of humus below the soil surface. Thus, the air-filled pore space, which results from the porosity minus the soil moisture, is on an annual average very similar in the top 10 cm and in the top 40 cm of the soil. Moreover, the air-filled pore space is not larger for the DWD data product than for GLDAS-Noah. Therefore, even with the high-resolution soil moisture and the vertically resolved porosity data from the DWD model, the $^{222}$Rn fluxes are still roughly 25% smaller compared to the a posteriori flux in the Germany domain. However, also the bottom-up $^{222}$Rn fluxes based on the DWD data show a higher (15%) temporal variability for the 0-10 cm soil moisture average than for the 0-40 cm average. Hence, the normalized standard deviation of the DWD-based $^{222}$Rn flux based on the top 10 cm soil moisture data is again similar to the normalized standard deviation of the a posteriori $^{222}$Rn flux.

## 4 Discussion

### 4.1 Does the $^{222}$Rn data coverage allow robust flux estimates in central Europe?

So far, inverse modelling has only rarely been used to estimate $^{222}$Rn fluxes. We are aware of a study by Hirao et al. (2010), who estimated the Asian $^{222}$Rn fluxes with a Bayesian inversion using atmospheric $^{222}$Rn observations from seven sites in East Asia and an Eulerian transport model. Their results indicate a higher $^{222}$Rn flux in East Asia than suggested by an a priori estimate from a $^{222}$Rn exhalation model. However, to our knowledge, a $^{222}$Rn inversion has not yet been performed over Europe. Therefore, we first investigated whether the current coverage of atmospheric $^{222}$Rn observations is sufficient to estimate robust $^{222}$Rn fluxes over Europe. We found that the annual mean and the temporal variability of the $^{222}$Rn flux estimates for the whole European model domain strongly depend on the prior information used. This can be explained by the poor coverage of available $^{222}$Rn observations in large parts of Europe. Currently, the Integrated Carbon Observation System (ICOS) atmospheric station network is planning to release hourly resolved $^{222}$Rn activity concentration measurements from several tower sites on a regular basis. Provided the number of ICOS sites carrying out $^{222}$Rn measurements increases in the future as was suggested previously (ICOS RI, 2020) this will improve the availability of $^{222}$Rn data in Europe.

Our inversion results are much less affected by the prior information if we restrict the domain to a region around Germany, a region well covered by $^{222}$Rn observations. For this region, the differences between the annual mean $^{222}$Rn flux estimates of various inversion configurations are much smaller (on the order of 10 %) compared to the annual mean differences between the process-based $^{222}$Rn fluxes (> 60 %). This indicates that the current data coverage in central Europe enables robust inversion

results that are mainly determined by the observational data and that are relatively independent of the choice of the a priori fluxes. This is a prerequisite for reliably evaluating process-based $^{222}$Rn fluxes with an atmospheric transport inversion.

The fact that our $^{222}$Rn observation data set cannot constrain the high-flux regions in Spain, Portugal or Italy could affect the inversion results for our regional target domain in central Europe, if fluxes from these regions are transported to Germany by advective winds. To assess this effect, we performed another inversion run using additional $^{222}$Rn observations from the mountain site Puy de Dôme (PUY), which is located in the southern part of France, where the process-based $^{222}$Rn flux maps indicate elevated fluxes, too (see Fig. 3). The impact of the PUY site on the annual mean $^{222}$Rn flux in the Germany domain is

less than 3% (see Appendix E). As the high-flux regions on the Iberian Peninsula and in Italy are further away from our target region than the PUY site, the effects of dilution and radioactive decay should be even larger. Therefore, we expect these high-flux regions to have only a minor influence on the $^{222}$Rn flux estimates for the Germany domain.

Finally, beyond the importance of good data coverage in the region of interest, differences in the scale of the $^{222}$Rn observations

directly translate into the inversion results and determine the extent to which absolute $^{222}$Rn fluxes can be estimated with a $^{222}$Rn inversion (see Fig. 9). This illustrates the need for further intercomparison projects between the $^{222}$Rn instruments as well as a well-calibrated and SI-traceable $^{222}$Rn data set. A proposed standardized protocol for the harmonization of $^{222}$Rn measurements has recently been published by Kikaj et al. (2025) thanks to the traceRadon project.

### 4.2 What is the performance of process-based European $^{222}$Rn flux maps?

Our inversion leads to ca. 20% (GLDAS) and almost 100% (ERA5) larger annual mean $^{222}$Rn fluxes than the process-based $^{222}$Rn fluxes in the Germany domain, and the temporal variability is also higher in the posterior flux. Estimating correct absolute $^{222}$Rn fluxes with a $^{222}$Rn inversion is challenging due to the aforementioned differences in the scale of the $^{222}$Rn observations as well as general uncertainties of atmospheric transport inversions, such as the representation of vertical mixing or lateral boundary conditions. In fact, we found that the FLEXPART and NAME transport models lead to up to 12% lower annual

mean $^{222}$Rn fluxes in comparison with STILT. In contrast, using the ANSTO scale instead of the HRM scale would lead to even higher $^{222}$Rn flux estimates, and thus further increase the bias compared to the process-based fluxes. The effect of the lateral boundary conditions on the annual mean $^{222}$Rn fluxes in central Europe is on the order of a few percent: Using alternative boundary conditions based on CAMS instead of TM3 results in 6% higher annual mean fluxes in the Germany domain, and using zero boundary conditions would lead to even 12% higher fluxes (see Fig. F1). From that we conclude that especially the

ERA5 $^{222}$Rn fluxes might be underestimated in central Europe, and that the bias compared to the posterior flux is unlikely to be fully explained by deficits in the inversion. This is consistent with other studies, which found that the ERA5 soil moisture might be too high in central Europe (Li et al., 2020; Karstens and Levin, 2024).

The significant (anti-) correlation between the posterior flux of the flat-prior inversion and the soil moisture data indicates that the temporal variability in the inversion signals are reliable and not only caused by inversion noise. Compared to GLDAS-Noah, the ERA5-Land soil moisture leads to a larger anti-correlation, indicating that it may better describe the temporal variability of the soil moisture in central Europe. This finding seems to be corroborated by a direct comparison with soil moisture measurements from several sites in Germany (see Fig. H1). Overall, the differences between the a priori and the a posteriori $^{222}$Rn fluxes might give a rough estimate for the uncertainty of the process-based $^{222}$Rn fluxes (see Fig. 9).

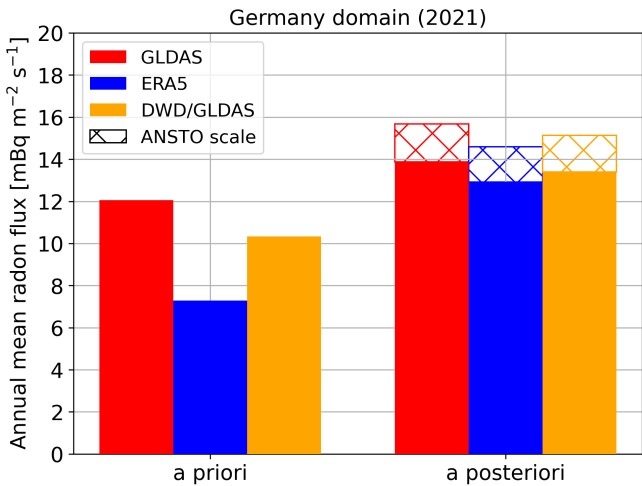

**Figure 9: Annual mean $^{222}$Rn fluxes in the Germany domain for 2021. Shown are the process-based GLDAS, ERA5, and DWD/GLDAS prior $^{222}$Rn fluxes, and the a posteriori fluxes based on the respective prior fluxes. The hatched range indicates the increase in the a posteriori fluxes when the ANSTO $^{222}$Rn observation scale is used instead of the HRM scale.**

### 4.3 What can we learn from the inversion results to improve the $^{222}$Rn flux maps?

The comparison between the temporal variability of the posterior $^{222}$Rn flux estimate of the flat-prior inversion and the soil moisture variability shows that soil moisture is an important driver for the temporal variability of the $^{222}$Rn fluxes and that the $^{222}$Rn flux variability is mainly driven by soil moisture in the top soil layer. The latter could be explained by the following points: (1) $^{222}$Rn gas produced in deeper soil layers has to travel a longer distance in the soil before it reaches the atmosphere and may be more affected by radioactive decay than $^{222}$Rn gas produced in upper soil layers. The soil moisture in the upper soil layers would therefore have a greater effect on the $^{222}$Rn flux at the surface than soil moisture in the deeper soil layers. (2) Depending on the type of soil, precipitation may take days to reach the deeper layers of the soil (cf. Fig. G1). As a result, air from the soil layers closer to the surface will outgas first, and the outgassing of air from deeper layers will be delayed. Soil moisture averaging over several layers may dilute the expected correlation between soil moisture and $^{222}$Rn flux at the surface. (3) Similarly, if the upper layer is saturated after smaller amounts of precipitation, but there are still many air-filled pores underneath, the top saturated layers reduce the air-filled pore space and therefore diffusion potential. $^{222}$Rn from deeper soil layers will hardly reach the surface.






The temporal variability of the process-based $^{222}$Rn fluxes increases for all three soil moisture products GLDAS, ERA5 and DWD when $^{222}$Rn fluxes are calculated using the soil moisture data from the top 10 cm instead of the top 40 cm of the soil column, resulting in $^{222}$Rn fluxes with similar temporal variability as our a posteriori flux. This finding could also open up the possibility of creating $^{222}$Rn flux maps using satellite-based soil moisture retrievals, which are only sensitive to the top few centimeters of soil. Even though such satellite data can have large gaps, e.g., due to snow cover, frozen ground or strong vegetation cover, such an application for improved $^{222}$Rn flux maps might be possible.

Absolute $^{222}$Rn fluxes are strongly dependent on the air-filled pore space, which determines the strength of the diffusion process. The comparison between GLDAS-Noah and ERA5-Land illustrates that an annual mean difference of 0.06 m$^3$/m$^3$ in the air-filled pore space already leads to a difference of more than 60 % in the annual mean $^{222}$Rn fluxes in the Germany domain. However, using the soil moisture and porosity data from the top 10 cm instead of the top 40 cm of soil has almost no effect on the annual mean $^{222}$Rn fluxes. This is because, in the case of GLDAS-Noah and ERA5-Land, the annual mean soil moisture is similar for the top 10 cm and the top 40 cm, and the porosity is vertically constant. The DWD soil moisture is higher in the top 10 cm than in the top 40 cm. However, this is compensated by the (vertically resolved) porosity, which is also higher in the top 10 cm than in the top 40 cm. Hence, the annual mean of the air-filled pore space and of the $^{222}$Rn flux are similar for the top 10 cm and top 40 cm soil data. If a vertically averaged porosity is assumed for the DWD data (i.e., if the average porosity of the top 40 cm is also used to calculate the air-filled pore space of the top 10 cm of soil), the air-filled pore space of the top 10 cm would be ca. 0.02 m$^3$ m$^{-3}$ smaller than that of the top 40 cm. Consequently, the resulting $^{222}$Rn fluxes based on the top 10 cm instead of the top 40 cm soil moisture would be on the order of 10-20% lower. Therefore, soil moisture impacts not only the temporal variability of the $^{222}$Rn fluxes, but together with porosity also the absolute $^{222}$Rn fluxes. Thus, an overestimation of the soil moisture and/or an underestimation of the porosity can easily change the absolute $^{222}$Rn fluxes. Overall, a reduction of the differences in the absolute values of different soil moisture and porosity products would further constrain process-based $^{222}$Rn fluxes. In this respect an expansion of soil moisture measurement networks such as the International Soil Moisture Network (ISMN), the Cosmic-ray soil moisture monitoring network (COSMOS) or similar would be desirable. Furthermore, cosmic-ray neutron sensing provides soil moisture data with a spatial representativeness of a few hundred meters (Köhli et al., 2015). This might make it better suited to evaluating (coarse-resolution) soil moisture reanalysis products, which can show significant differences compared to local (point) measurements (cf. Fig. H1).

Of course, there may also be other parameters or processes that are not or only inadequately described in the $^{222}$Rn flux model and could explain a bias in the absolute fluxes: e.g., an underestimation of the $^{222}$Rn source due to too low radium concentrations or a too low emanation coefficient. In addition, advective fluxes, e.g. induced by a pressure gradient between the atmosphere and the soil, could lead to a $^{222}$Rn flux contribution. However, their overall contribution might be negligible compared to the diffusive flux because the permeability of the soil is several orders of magnitude smaller than the diffusion coefficient, and the hydrostatic pressure gradients are typically small (López-Coto et al., 2013; Nazaroff, 1992). Nevertheless,

changes in atmospheric pressure e.g. due to weather fronts can induce short-term variability in the $^{222}$Rn flux: decreasing atmospheric pressure is expected to suck $^{222}$Rn-rich air from the soil into the atmosphere, whereas rising atmospheric pressure forces $^{222}$Rn-poor air from the atmosphere into the soil (Clements and Wilkening, 1974). However, due to the compensating effects of decreasing and rising atmospheric pressure, the overall effect of pressure is larger on instantaneous $^{222}$Rn flux variability than on absolute (time-averaged) $^{222}$Rn fluxes (Schery et al., 1984; Ishimori et al., 2013).

Indeed, we found no significant temporal correlation ($R^2 < 0.05$) between the a posteriori $^{222}$Rn flux and the atmospheric pressure (and its first derivative) in the Germany domain and also for a smaller domain in south-west Germany. This may indicate that the $^{222}$Rn flux variability is much more influenced by soil moisture than by pressure changes on the time scale accessible to the inversion (i.e. daily resolution) and/or that the pressure effect is superimposed by other meteorological effects (e.g. wind, precipitation). Overall, our inversion results provide information on $^{222}$Rn exhalation processes occurring on a daily or longer time scale. The processes driving sub-daily $^{222}$Rn flux variability, which can be on the order of 20–50% (Rábago et al, 2022), may be difficult to investigate using our inversion results.

### 4.4 Outlook

Overall, further development of the $^{222}$Rn flux maps is needed. Our inversion results could already give some indications on how to further improve the maps. It would also be interesting to compare our inversion results with other $^{222}$Rn flux maps that are based on different $^{222}$Rn flux models or even completely different methods such as $^{222}$Rn flux estimation using the terrestrial γ-dose rate (Szegvary et al., 2009). While Szegvary et al. (2009) presented a 'static' inventory based on specific and punctual measurements, continuous γ-dose rate time series, e.g. from the European Radiological Data Exchange Platform (EURDEP), may enable time-resolved γ-dose rate based $^{222}$Rn flux maps. These maps could then be used alongside our inversion results to further investigate temporal variability in $^{222}$Rn fluxes.

Finally, accurate $^{222}$Rn flux maps are important not only for evaluating atmospheric transport models or constraining the emissions of other trace gases, e.g., using the Radon-Tracer Method (RTM, e.g., Grossi et al., 2018; Levin et al., 2021; Curcoll et al., 2024). They are also relevant to the radiation protection community, because $^{222}$Rn is one of the major sources of natural radiation to which the population is exposed. Nowadays, European countries are developing national plans for achieving their radiation protection goals. These plans need to count on reliable $^{222}$Rn flux maps with a high spatial and temporal resolution for assessing so-called radon priority areas (RPAs; Röttger et al., 2021). Therefore, the evaluation of $^{222}$Rn flux maps using a $^{222}$Rn inversion might be of particular interest to both the climate and the radon protection community.

# 5 Conclusions

The characteristics of [222]Rn make it a powerful natural tracer for atmospheric transport that can be used to validate and calibrate atmospheric transport models or to estimate GHG fluxes, e.g., with the Radon-Tracer Method (RTM, e.g., Levin et al., 2021).
However, all these applications require an accurate estimate of the [222]Rn flux. So far, [222]Rn fluxes have mainly been evaluated by local [222]Rn flux measurements, which have a limited spatial representativeness. In this study, we investigated the potential of a top-down approach to evaluate process-based [222]Rn fluxes, although we are well aware that such an approach is cyclic and assumes unbiased model transport, which of course may not be the case (otherwise the information from [222]Rn observations would no longer be needed). We carefully assessed the impact of potential deficits in the transport models by performing
several inversion runs with three different transport models. We found that the current coverage with [222]Rn observations in Europe allows robust and data-driven [222]Rn flux estimates within a region covering Germany, which also provide indications on how to improve the process-based [222]Rn flux maps.

We evaluated two process-based [222]Rn flux maps for Europe, which are based on two different soil moisture reanalyses
products (GLDAS-Noah versus ERA5-Land). Our a posteriori annual mean flux of about $14 \pm 4$ mBq m$^{-2}$ s$^{-1}$ (mean $\pm$ std) is ca. 20% (GLDAS) and almost 100% (ERA5) higher than the process-based [222]Rn fluxes in a domain covering Germany in 2021. Furthermore, both [222]Rn flux maps tend to underestimate the temporal variability of the [222]Rn fluxes, although the variability of the ERA5-based [222]Rn fluxes is in better agreement with the variability of our inversion results.

We found a significant correlation of r=-0.6 and r=-0.8 between the posterior flux of a flat-prior inversion, which is independent of any prior information about spatial and temporal flux variations, and the GLDAS-Noah and ERA5-Land soil moisture, respectively. Moreover, the soil moisture time series lag a few days behind the [222]Rn flux if the soil moisture is averaged over a too large depth (i.e. > 40 cm). In contrast, the time series of the soil moisture and the [222]Rn flux are time-synchronous if the soil moisture average of the top 10 cm of soil is used. This indicates that the temporal [222]Rn flux variability is mainly caused
by the soil moisture variability in the top 10 cm of soil. Indeed, we were able to increase the temporal variability of the process-based [222]Rn fluxes by using soil moisture data from the top 10 cm only, resulting in a temporal variability similar to that of the posterior flux estimate.

Finally, realistic uncertainty estimates for the [222]Rn flux maps are required when [222]Rn is used in joint inversions for a targeted
tracer such as methane (CH$_4$). Such a dual-tracer inversion directly incorporates the [222]Rn information on the transport model performance by exploiting the fact that the transport model error (as part of the model-data mismatch error) is correlated between the targeted tracer (e.g., CH$_4$) and [222]Rn. In a subsequent study, we will investigate whether this information can help to improve the top-down flux estimates of the targeted tracer (CH$_4$).

## Appendix

**A How to average the $^{222}$Rn fluxes for the time-aggregated FLEXPART and NAME footprints?**

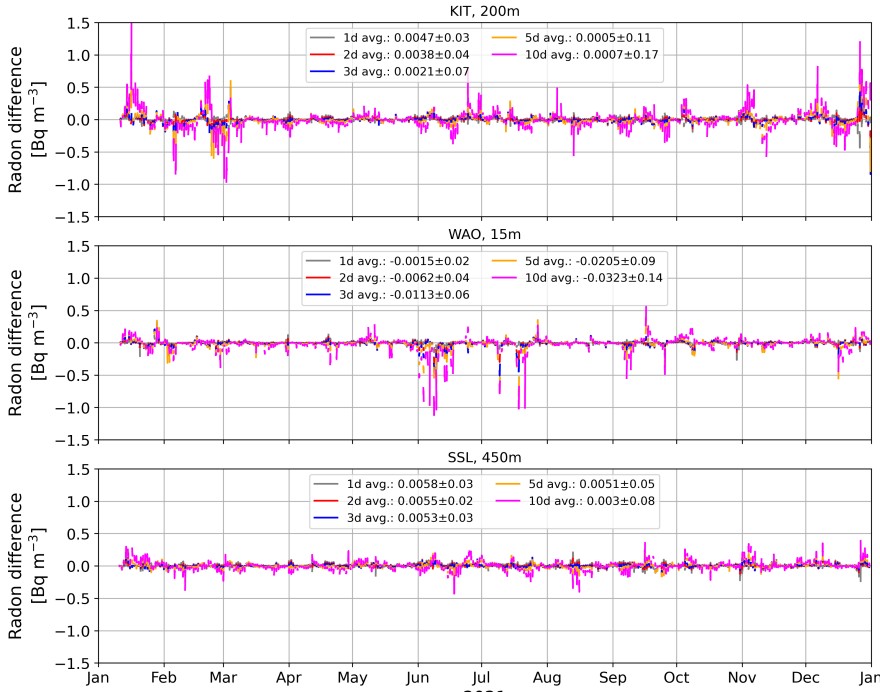

**Figure A1: Modelled $^{222}$Rn concentration differences between the simulations based on time-aggregated and time-resolved STILT footprints. The time-resolved STILT footprints are mapped with daily $^{222}$Rn fluxes (based on GLDAS-Noah soil moisture). The time-aggregated STILT footprints are mapped with GLDAS $^{222}$Rn fluxes, which were averaged over 1 (grey), 2 (red), 3 (blue), 5 (orange), and 10 days (magenta), respectively. This experiment has been performed to deduce the most appropriate time interval over which the daily $^{222}$Rn fluxes should be averaged when they are mapped with the time-aggregated FLEXPART and NAME footprints. The results are shown for a continental (KIT200), a coastal (WAO15), and a mountain (SSL450) site. From this study we found that averaging the $^{222}$Rn fluxes over 3 days might lead to a good compromise between low bias and low standard deviation between the time-aggregated and time-resolved STILT runs (see Sect. 2.3).**

**B Fits to observations**

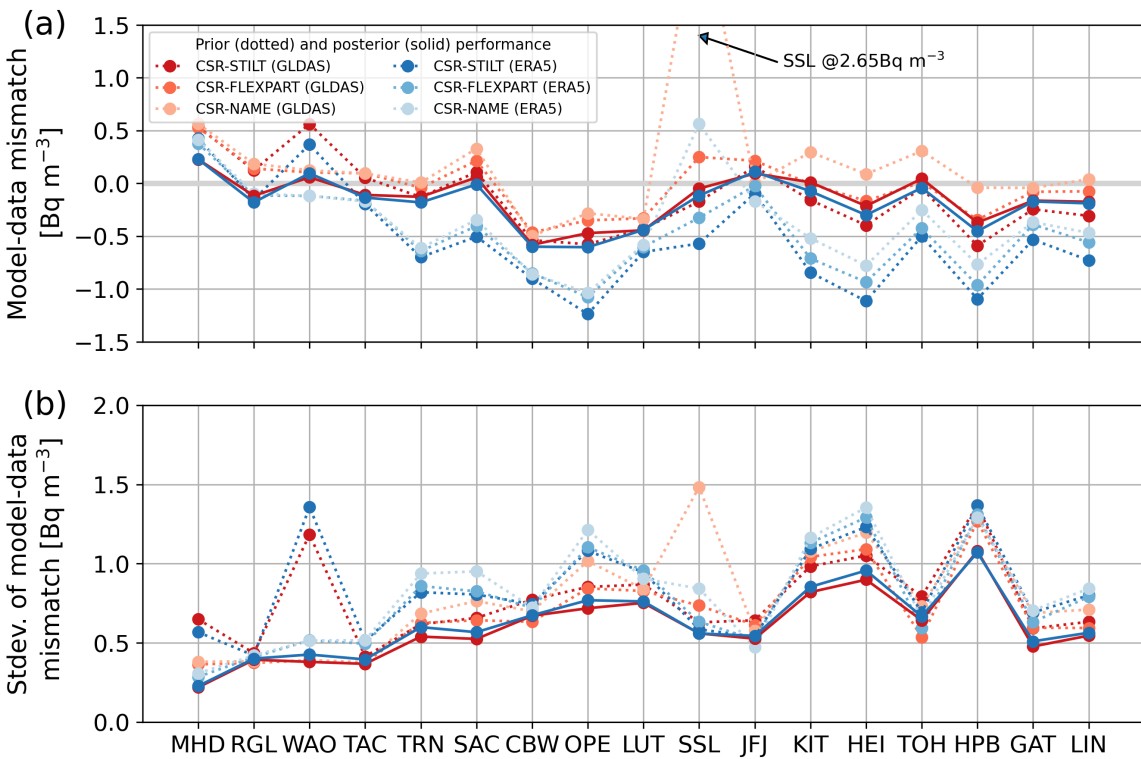

**Figure B1: Fits to the observations for the CSR-STILT, CSR-FLEXPART, and CSR-NAME setups, using the GLDAS and ERA5 a priori fluxes. Shown are annual mean model-data mismatch (a) and its standard deviation (b) for the a priori (dotted) and the a posteriori fluxes (solid) for each observation site.**


**C Impact of the observations from Cabauw (CBW)**

As a leak in the $^{222}$Rn inlet line at the CBW site may have contaminated the $^{222}$Rn measurements in 2021 at this site, we investigated the impact of the CBW observations on the inversion results. For this, we performed an additional (flat-prior)

inversion run without using the observations from CBW. The resulting posterior $^{222}$Rn fluxes are a few percent lower in the surroundings of CBW if the CBW observations are not used. However, averaged over the Germany domain, the annual mean flux is only less than 1% lower.

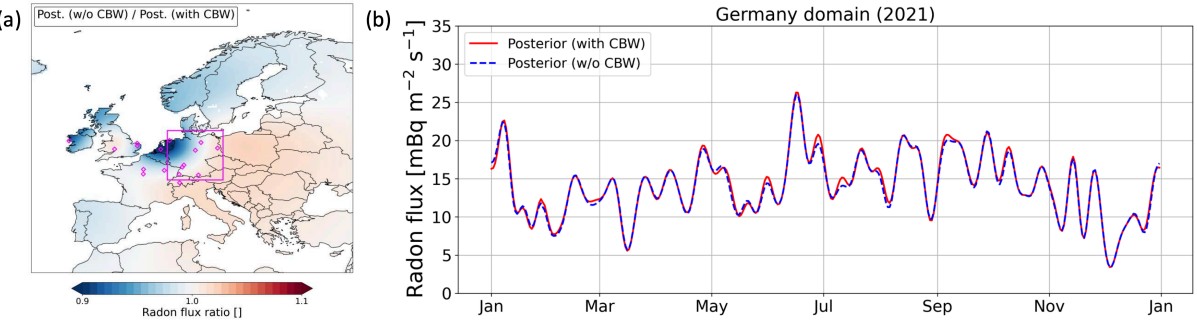

**Figure C1: Impact of the observations from Cabauw (CBW) on the inversion results. (a) Ratio between the [222]Rn posterior flux of the inversion without CBW data and the standard inversion with CBW data. (b) Time series for the full year of both posterior [222]Rn fluxes in the Germany domain. D Seasonal posterior-prior [222]Rn flux differences**

## D Seasonal posterior-prior $^{222}$Rn flux differences

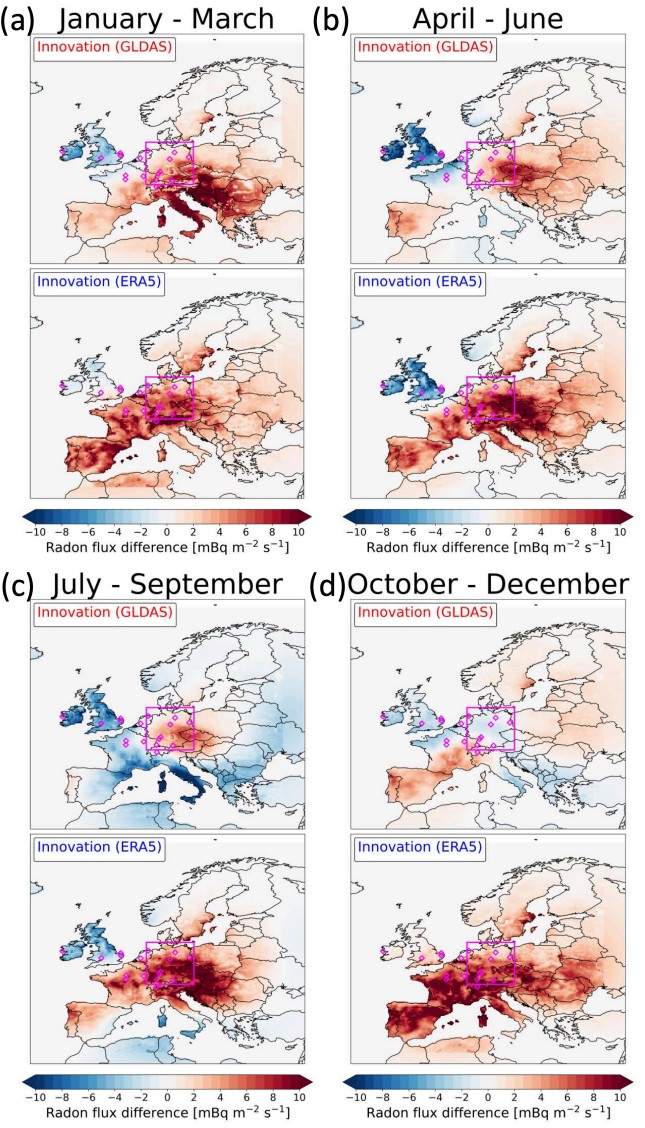


**Figure D1: Innovation (posterior-prior $^{222}$Rn flux differences) for CSR-STILT inversion runs using the GLDAS (red) and ERA5 (blue) prior fluxes, averaged for the first (a), second (b), third (c), and forth (d) quarter of 2021.**

## E Impact of unconstrained high-flux regions on inversion results in the Germany domain

To assess the impact of distant high-flux regions not covered by our observations on the inversion results in the Germany

domain, we made a sensitivity inversion run using the additional observations from the mountain site Puy de Dôme (PUY).

This site is located in the southern part of France, for which the process-based $^{222}$Rn flux maps indicate elevated $^{222}$Rn fluxes (see Fig. 3).

We initially excluded the PUY site in our inversion system due to the following reasons: The PUY site uses a LSCE $^{222}$Rn instrument, and there has never been made an intercomparison between the LSCE and the HRM $^{222}$Rn instruments at that specific mountain site. Therefore, it is unclear how to scale the PUY measurements to make them comparable to the HRM scale. Furthermore, the PUY site is challenging to represent in the transport model due to its very steep terrain.

Nevertheless, for this PUY inversion run, we assume a scaling factor of 0.7 to make the PUY measurements comparable to the

HRM scale. This factor has been found in Schmithüsen et al. (2017) for the French site Gif-sur-Yvette (GIF), which has a similar $^{222}$Rn intake height above ground as the PUY site. However, while the GIF site is located 167 m asl, the PUY site is located 1465 m asl, which calls into question the suitability of the assumed scaling factor. Therefore, we will only interpret the PUY inversion results in a qualitative way.

Figure E1 shows the posterior flux difference between the PUY inversion, which uses the additional observations from the PUY site, and our standard inversion (without PUY). The largest flux differences occur southwest of the PUY site in the southern part of France and on the Iberian Peninsula, where our base inversion is poorly constrained due to the lack of observations. The south-westerly orientation of the footprint of the PUY site can be explained by the predominant influence of south-westerly winds in that region. Consequently, the PUY observations contain mainly information on the fluxes in south-

west Europe. Indeed, the impact of the PUY observations on the fluxes in the Germany domain are small. When including the PUY observations, the annual mean flux in the Germany domain changes by less than 3%.

From that, we expect that the potential influence from high-flux regions in south-west Europe, e.g. in Spain, Portugal and Italy, on the inversion results in the Germany domain is negligible. Those regions are even more distant than the PUY site, and the

effect of dilution and radioactive decay should therefore be even larger.

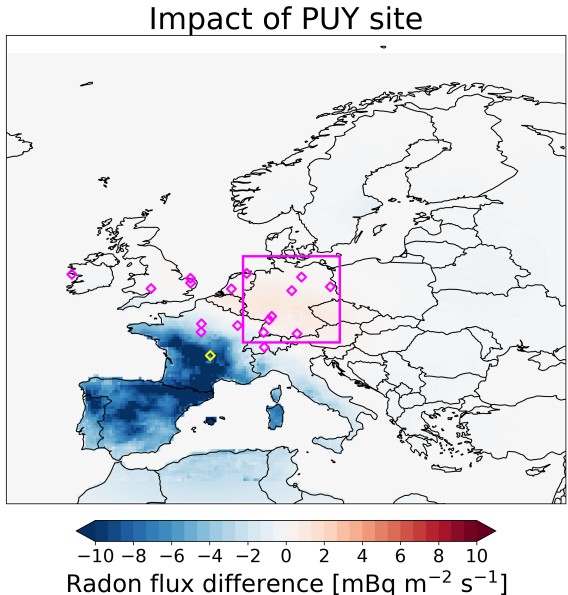

**Figure E1: Posterior** $^{222}$**Rn flux difference between an inversion run with and without using the** $^{222}$**Rn observations from the PUY site (marked in yellow). Both inversions use the GLDAS prior fluxes.**

## F Impact of $^{222}$Rn background on inversion results in Germany domain

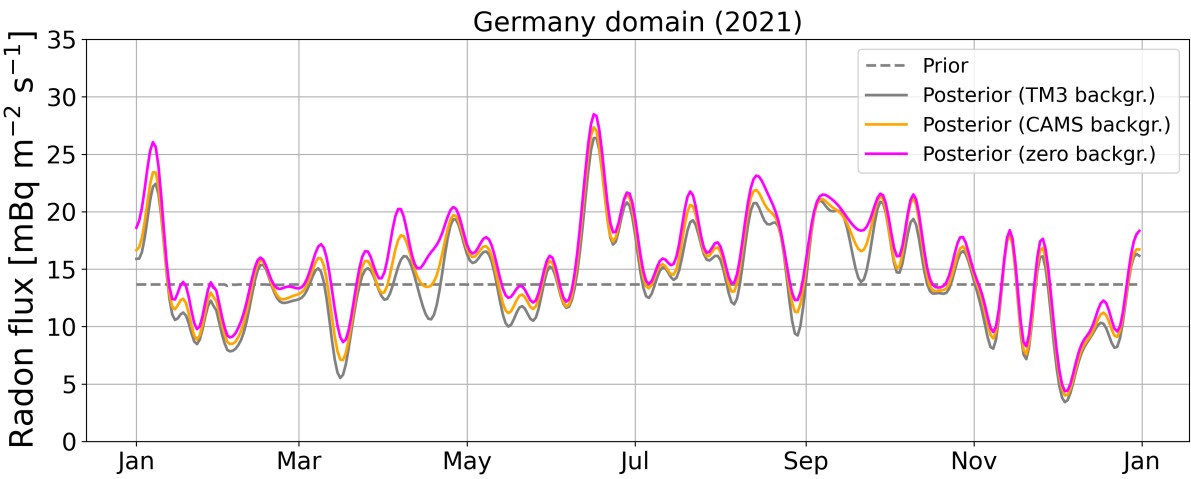


**Figure F1: Impact of the European** $^{222}$**Rn background on the** $^{222}$**Rn flux estimates for the Germany domain. Shown are the results of flat-prior inversions using a TM3-simulated background (solid grey curve), a CAMS (EAC4)-based background (orange), and a zero background (magenta).**

**G Difference between [222]Rn fluxes based on 0-10 cm and 0-40 cm soil moisture averages**

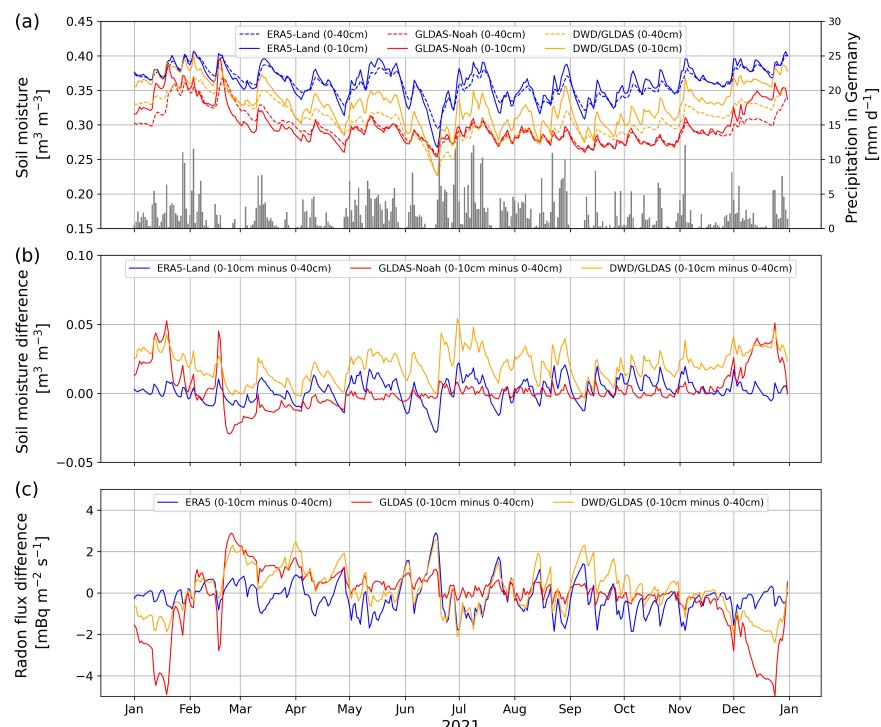


**Figure G1: (a) Mean ERA5-Land (blue), GLDAS-Noah (red) and DWD/GLDAS (yellow) soil moisture in the Germany domain, averaged between 0-10 cm (bold line) and 0-40 cm (dashed line), and mean precipitation in Germany (grey histograms). Note, that the soil moisture average corresponds to a slightly larger area (Germany domain) than the precipitation average (Germany only, data taken from DWD, 2024). (b) Difference between the soil moisture 0-10 cm and 0-40 cm averages for the Germany domain. (c)**
**Differences between the [222]Rn fluxes based on 0-10 cm and 0-40 cm soil moisture data.**

# H Comparison of soil moisture reanalysis products with observations

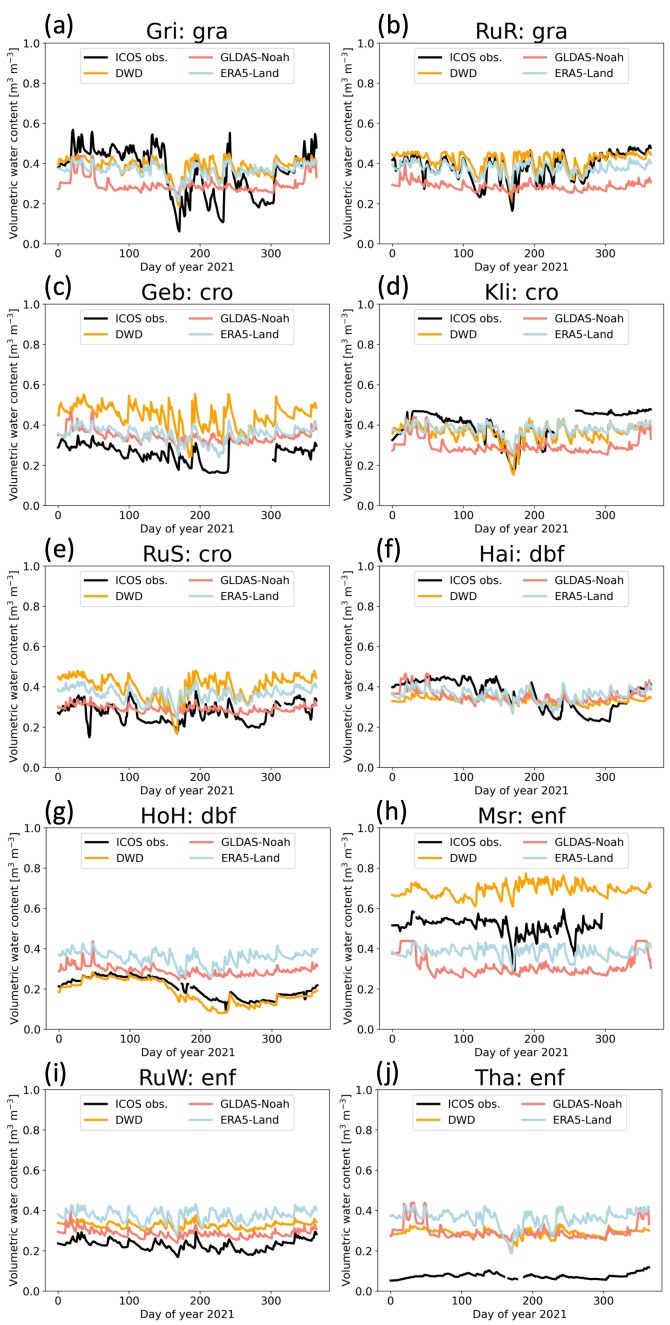

**Figure H1: Comparison between the GLDAS-Noah (red), ERA5-Land (blue) and DWD (orange) soil moisture with measurements (black, data taken from ICOS RI, 2024) performed at different ICOS ecosystem sites in Germany. The modelled soil moisture shows the average between 0-10 cm, and the measurements show the mean soil moisture from all sensors within the 0-10 cm soil layer. The first three-letter code of the subplot titles indicates the measurement site, and the second three-letter code refers to the site class: gra – grasslands, cro – croplands, dbf – deciduous broadleaf forests, enf – evergreen needleleaf forests.**


## Code and data availability

The process-based $^{222}$Rn flux maps from Karstens and Levin (2022a,b) can be accessed from https://hdl.handle.net/11676/JoDR653JxQuqLvEwzqI2kdMw and https://hdl.handle.net/11676/NvC7D-BVXlnHtFBdUSKpNVHT. The alternative $^{222}$Rn flux maps based on the DWD soil moisture, along with the different posterior $^{222}$Rn flux estimates, can be found in Maier (2025). The HRM $^{222}$Rn data from the sites of the German Weather Service (DWD) are compiled in Fischer et al. (2024) and are publicly available at https://doi.org/10.18160/Q2M8-B1HJ. The ANSTO $^{222}$Rn

data from JFJ, OPE, SAC, TRN and WAO can be downloaded from the Data Portal of the ICOS Carbon Portal (https://data.icos-cp.eu/portal, Emmenegger et al. (2021), Ramonet et al. (2021a,b,c), Forster and Manning (2021)). The ANSTO $^{222}$Rn data from RGL and TAC are available through the Centre for Environmental Data Analysis (CEDA) at https://catalogue.ceda.ac.uk/uuid/bd7164851bcc491b912f9d650fcf7981 (O'Doherty et al., 2024).

## Author contributions

FM, CG, IL and UK designed this study of a $^{222}$Rn inversion in Europe. CG, IL, UK, CR and MG provided valuable suggestions during inspiring discussions. FM, CR and MG implemented $^{222}$Rn as a tracer in the CarboScope-Regional inversion framework. UK and IL developed the $^{222}$Rn flux model and UK and FM calculated process-based $^{222}$Rn fluxes using this model. EF provided high resolution soil moisture and porosity data for Germany used in the $^{222}$Rn flux model. MG corrected and compiled the $^{222}$Rn data from the HRM instruments and DK evaluated and provided the ANSTO $^{222}$Rn data from the United Kingdom.

IL and FM harmonized the European $^{222}$Rn dataset. SH and AM calculated the FLEXPART and NAME footprints, respectively. MG simulated the background $^{222}$Rn concentrations. FM performed the $^{222}$Rn inversion runs and wrote the manuscript with the help and input of all co-authors.

## Competing interests

At least one of the (co-)authors is a member of the editorial board of Atmospheric Chemistry and Physics.

## Acknowledgements

We would like to thank Theresia Yazbeck for the internal review of this manuscript and her valuable comments. We acknowledge the Integrated Carbon Observation System (ICOS), the German Weather Service (DWD), the University of Bristol and the Heidelberg University for performing $^{222}$Rn activity concentration measurements and providing the data. The $^{222}$Rn data from LUT and CBW were evaluated and provided by Hubertus A. Scheeren and Arnoud Frumau, respectively.

Furthermore, we thank Victor Kazan and Michel Ramonet from LSCE for providing the $^{222}$Rn data from MHD. We are grateful to Marco Liedtke and Paul Schmidt-Walter from the DWD for generating high resolution soil moisture and porosity maps for Germany. We thank the following PIs of the German ICOS Ecosystem stations (Christian Brümmer (DE-Geb), Matthias Mauder (DE-Gri, DE-Kli, DE-Tha), Alexander Knohl (DE-Hai), Anke Hildebrandt (DE-HoH), Matthias Drösler (DE-Msr), Marius Schmidt (DE-RuR, DE-RuS, DE-RuW) for providing soil moisture data. Special thanks go to the ATM (Airborne trace gas measurements and mesoscale modelling) group at the Max Planck Institute for Biogeochemistry for the stimulating discussions and their support. This work used resources of the Deutsches Klimarechenzentrum (DKRZ) granted by its Scientific Steering Committee (WLA) under project ID bm1400.

**Financial support**

This work is funded by the German Federal Ministry of Education and Research (BMBF) project Integrated Greenhouse Gas Monitoring System for Germany (ITMS)- Modelling Module under grant number 01LK2102A.

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
