# Peer review of "How reliable are process-based 222radon emission maps? Results from an atmospheric 222radon inversion in Europe"

_EGUsphere, 2025_

## Referee Comment (RC2)

The manuscript evaluates the reliability of process-based $^{222}$Rn emissions in Europe using atmospheric inversion modeling—an important and timely topic within the scope of Atmospheric Chemistry and Physics (ACP). By assessing different soil moisture-driven emission maps through an inversion framework constrained by $^{222}$Rn observations, the study provides insights into limitations of current $^{222}$Rn emission modeling approaches.

Major Concerns:

Writing and structure: The writing quality needs some improvement. Several section titles are overly informal, and some figure captions are too lengthy or imprecise. Enhancing the clarity, consistency, and scientific tone of the manuscript would significantly improve its accessibility and impact.

Methodological clarity: There are a number of unresolved issues in the methodological approach, particularly regarding the inversion framework. These include inconsistencies in particle counts across models, the interpretation of time-aggregated versus time-resolved footprints, the background subtraction procedure, and the handling of prior–posterior dependencies. Some aspects of the inversion should be more rigorously justified or supported with sensitivity analyses.

Limited broader context: The manuscript would benefit from a broader discussion of the implications of its findings—particularly regarding how the improved $^{222}$Rn flux estimates might support applications beyond radon itself, such as in evaluating atmospheric transport schemes or constraining emissions of other trace gases.

Strengths:

Despite these concerns, the manuscript includes a number of thoughtful discussions.

The authors critically assess the data coverage of $^{222}$Rn observations, highlighting how observational density shapes the inversion's dependence on prior information.

The manuscript provides a breakdown of uncertainty sources, including observational scaling, model assumptions, and soil parameter variability.

The soil moisture sensitivity analysis, particularly the exploration of top-10 cm versus deeper layer effects, is a valuable contribution and has potential implications for improving future radon flux parameterizations.

Line by line comments:

Line 17: In the abstract, "In this study, we evaluate two process-based 222Rn flux maps for Europe based on two different soil moisture reanalysis products (GLDAS-Noah and ERA5-Land) using the flux results obtained from a one-year 222Rn inversion performed with the CarboScope-Regional inversion system and 222Rn observations from 17 European sites." The sentence is quite long, making it challenging to follow on a first read. My suggestion is but may not be accurate:

"This study evaluates two process-based $^{222}$Rn flux maps for Europe. The maps are developed separately using different soil moisture reanalysis products, GLDAS-Noah and ERA5-Land. The evaluation is based on flux results from a one-year $^{222}$Rn inversion, which was conducted using the CarboScope-Regional inversion system, and observational data from 17 European sites."

What is the role of the inversion system? Does it used to estimate surface 222Rn concentrations using the two soil moisture reanalysis products?

Line 138: "Whilst the GLDAS-Noah and ERA5-Land models assume that the soil porosity is constant over the entire soil column, AMBAV uses vertically resolved soil porosity data (from Hartmann et al., 2024)."

What could be the magnitude of the impact from using constant soil porosity (as in GLDAS-Noah and ERA5-Land) versus vertically resolved porosity (as in AMBAV)? Can this help explain the comparisons later in the manuscript?

Line ~150, Table 1: It would be better to expand Table 1 to include additional information such as temporal resolution, type of soil model used, and porosity representation (e.g., 3-hourly, constant vs. vertically resolved). This would help readers understand their influence on the $^{222}$Rn flux calculations.

Line 155:Title of Section 2.2 is the same as that of Section 2.1. It could be renamed "$^{222}$Rn Observations" or something similar.

Line ~215: The caption of Table 2 is not sufficiently concise. Some information currently in the caption could be moved to the main text or directly represented in the table layout.

Latitude coordinates should explicitly indicate N (North) or S (South) to avoid ambiguity.

Consider rephrasing the caption for clarity and consistency. For example:

"Table 2. Overview of the 17 European sites providing atmospheric $^{222}$Rn measurements."

Also consider including symbol definitions as footnotes directly under the table, such as:

T – Tower sites

* – High-altitude sites

Line ~227: The manuscript states that particles are released at regular hourly intervals rather than matched to the exact $^{222}$Rn measurement times. Any reason that the particles cannot be released at the exact measurement time?

Line ~230: It would be better to include an equation indicating how $^{222}$Rn concentrations are calculated from the footprint and fluxes. This would clarify the convolution process and demonstrate how radioactive decay is incorporated.

Line 235–240:The manuscript notes that 100 particles are released in STILT, compared to 20,000 in FLEXPART. This discrepancy raises concern. After 10 days of backward transport, especially within the boundary layer, the number of STILT particles in the footprint layer may be very small, potentially causing biased flux estimates. Could the authors clarify why such a low number is used in STILT? would increasing the number of STILT particles (e.g., to match FLEXPART) improve consistency and reduce sampling error?

Line 235–245: The authors use 30-day back-trajectories for FLEXPART and NAME, but only 10 days for STILT. Given the $^{222}$Rn half-life of 3.8 days, the contribution from emissions beyond ~10 days should be minimal. Could the authors clarify whether the 30-day simulations were necessary for resolving $^{222}$Rn fluxes?

Line ~245–255: FLEXPART and NAME use $^{222}$Rn-decay-corrected footprint products aggregated over 30 days, and that these footprints lack temporal resolution. Without knowing when the footprints are most sensitive, it is unclear how the daily fluxes can be meaningfully convolved with the aggregated footprints.

Line ~255–260: The authors justify their use of 3-day averaged fluxes with FLEXPART and NAME based on timing sensitivity inferred from STILT. However, STILT uses only 100 particles, which may not be sufficient to robustly resolve the timing distribution after 3–10 days of transport. Moreover, FLEXPART is capable of outputting time-resolved footprint

information. There are critical compromises in the intercomparison of the three model products.

Line ~260–275: The manuscript applies a modeled $^{222}$Rn background correction by subtracting TM3-simulated concentrations based on a globally uniform $^{222}$Rn flux map. However, this approach raises several concerns. First, the short lifetime of $^{222}$Rn means that long-range contributions to observed concentrations are inherently small. Subtracting a modeled background may introduce biases and artificially reduce the observational constraint. I do not think this is necessary. Second, the assumed constant fluxes outside Europe lack empirical justification and do not account for known variability in soil radium content, land cover, or meteorology. This can introduce noise in the seasonal/spatial variations.

Line ~280–285 (in the inversion system description): Including an equation of the inversion system would clarify how the posterior fluxes are derived and how prior information and observational constraints are weighted. A simple expression such as

$$J(x) = (x - x_{\text{prior}})^T \mathbf{B}^{-1}(x - x_{\text{prior}}) + (y - Hx)^T \mathbf{R}^{-1}(y - Hx)$$

along with brief definitions of terms.

x: unknown posterior $^{222}$Rn flux field; x prior: prior flux map

y: observed $^{222}$Rn concentrations; H: footprint-convolved fluxes

This may raise another critical issue. If my expression is correct, then H is NOT independent from x prior. It is calculated based on the flux map, which is x prior here. I am not sure if the inversion model is still valid if H is a dependent of x prior. Again, my expression can be wrong, so it is important for the authors to provide such an expression.

Line ~310, Section Title: The title "Temporal correlation between soil moisture and 222Rn flux variability" sounds like it will present a data-driven correlation study between two independently measured quantities: soil moisture and $^{222}$Rn flux variability. In fact, it investigates the impact of using a flat prior compared to the soil moisture based prior. I believe that this involves not only temporal variations but also spatial variations. The title can be something like "2.4 The impact of using Soil Moisture based $^{222}$Rn fluxes compared to a flat prior on the inversion system". This is probably not good enough. I leave it here for the authors to polish more.

Line ~360, Figure 3: The maps present annual means. However, given the high temporal variability of $^{222}$Rn fluxes driven by soil moisture and meteorology (as shown in the time

series), the maps cannot provide much information. It would strengthen the evaluation to also show the innovations at monthly or seasonal resolution.

Line ~370: Figure 3 shows that posterior $^{222}$Rn fluxes over the European domain exhibit a clear seasonal cycle, while the Germany-only domain shows much weaker seasonal variability, particularly in the flat-prior inversion. It would be better if the authors can discuss this difference and its implications for interpreting regional versus continental-scale flux patterns.

Line ~378: A more precise title could be "Sensitivity of Inversion Results to Model Configuration"

Line ~402: "The NAME forward simulations" have never been introduced before this point.

Line ~417: "However, there seems to be a slight seasonal cycle in the difference". Again, it would be better to have some discussion about reasons for such seasonal cycles.

Line ~448: The title "Alternative $^{222}$Rn flux maps" does not clearly reflect the scope of the section, which focuses on analyzing the temporal variability of posterior $^{222}$Rn fluxes and their correlation with soil moisture at different depths. Try from "Evaluating Soil Depth Sensitivity in $^{222}$Rn Flux".

Line ~475: The authors state that the annual mean flux is only marginally affected (2–3%). It would be valuable to also assess the impact on daily or monthly mean fluxes. Such resolution is particularly relevant for interpreting short-term atmospheric $^{222}$Rn observations and improving result fidelity.

Line ~516: Although I had a few concerns about the methodology, the discussions here are quite thoughtful. The sources of the big difference between process-based fluxes and the posterior fluxes are clearly articulated, and the uncertainty due to model assumptions versus prior biases is discussed carefully.

---

## Author Comment (AC1)

We would like to thank Claudia Grossi and the anonymous referee for their careful review of our manuscript and for their helpful and inspiring comments. Our responses to these comments are marked in blue. The line numbers correspond to those in the manuscript version with track changes.

**Response to review 1**

**Review of the manuscript egusphere-2025-477: How reliable are process-based 222radon emission maps? Results from an atmospheric 222radon inversion in Europe**

Within the manuscript egusphere-2025-477 authors make an effort to evaluate two process-based 222Rn flux maps for Europe based on two different soil moisture reanalysis products (GLDAS-Noah and ERA5-Land), available thanks to the results of the European traceRadon project, using the flux results obtained from a one-year (2011) 222Rn inversion performed with the CarboScope-Regional inversion system and hourly 222Rn observations from 17 European sites mainly located in central Europe.

The present study and its possible highlights and results, from my point of view, is of high scientific interest because the radioactive noble gas radon (222Rn) is a suitable tracer for atmospheric transport and mixing processes that can be used to evaluate and calibrate atmospheric transport models or to estimate greenhouse gas emissions using the so-called Radon-Tracer method (Grossi et al., 2018 and Curcoll et al., 2024). In addition, radon is one of the major source of natural radiation for the exposed population and nowadays European countries are developing radon national plans for radiation protection goals. These cited plans need to count on reliable, high spatial and high temporal resolution radon flux maps for assessing radon priority areas.

Thank you for highlighting radiation protection as another motivation for our study. We included it in the Discussions section of our manuscript (l. 723-727). We also added the references to the two studies on the Radon-Tracer method (l. 721-723).

For all these previous reasons I think that this manuscript is worthy of being presented to the scientific community. However, I would like to suggest to the authors of working on some specific aspects of the manuscript and its presentation with the main aim of improving it.

Please take into account that my comments and/or suggestions are exclusively based my experience related with the metrology of atmospheric concentration and fluxes of GHG as well as radon gases. Although I have some experience related with RTM applications and atmospheric transport models, I have no experience in inversion modelling.

**General and specific comments:**

**Abstract:** Authors may clarify that both radon activity concentration data and radon flux maps have been used only for 2011. Also is important to specify in the abstract that the 17 atmospheric sites are located only in central Europe. Finally, the names of the three transport models maybe cited in the text.

We revised the abstract and clarified that we only investigate the year 2021, and that the 17 atmospheric sites are located in central Europe (l. 17-21). We also added the names of the three transport models (l. 30).

**Introduction**

**Lines 46:** you could cite Galmarini, 2006 and Arnold et al., 2010

Thank you for these references. We included them (l. 49-50).

**Lines 52:** the work of Karstens and Levin, 2024 was not published within a peer-review journal. It appears as a report and it may be specified. Actually in this report the comparison between modeled and measured radon flux was carried out only at one site and during 6 months. This may be clarified within the text together with the need of the availability of long-term continuous and reliable radon flux measurements over Europe for validation studies.

We specified that the work of Karstens and Levin (2024) is a technical report, and that they only use continuous flux measurements from one site and during 6 months. We now also mention the need of long-term continuous measurements over Europe (l. 64-68). However, these measurements should have a high spatial representativeness so that they can be used to validate continental flux maps.

**Lines 66-73:** In this paragraph I miss the explanation of how the lack of atmospheric data over the area of interest (in this case Europe) may affect the absolute values and the uncertainties related with the inversion model results. It could be interesting adding a paragraph on the state of the art of the research on this aspect not only for radon. I insist on this aspect because in agreement with the Figure 4 (below) of Rottger et al., 2021 there are 31 atmospheric stations in Europe measuring radon concentration outdoor. In the case of Spain for example, Grossi et al., 2016 presented several atmospheric radon station data and I do not know if authors tried to contact the IP of these stations in order to know if 2011 data was available too. Spain as well as Italy count with really high radon fluxes which may influence atmospheric radon observed in central Europe.

[Figure]

**Figure 4.** Location of the 31 European sites measuring atmospheric radon measurements. 19 sites are part of ICOS Atmosphere (red circles). The others in blue circles are either part of AGAGE, WMO-GAW or are not affiliated to a GHG network. OpenStreetMap contributors 2020. Distributed under a Creative Commons BY-SA License. Accessed 11 June 2021.

Thank you for this comment. We are well aware that there is a lack of atmospheric radon observations in Europe, especially in the northern, eastern, and south-eastern part of Europe. This leads to a strong dependence of the European inversion results on the prior fluxes (see Fig. 4c in our manuscript). Therefore, we do not aim to evaluate the radon fluxes in the whole European domain. Indeed, we only analyse the inversion results in central Europe within a domain surrounding Germany (i.e., our area of interest), which is well covered by observational data.

For our study, we use the radon observations data set, which has been compiled for the traceRadon project. Unfortunately, for the year of our study (2021) there are only radon observations from one Spanish site (INTE), and only between mid-May and the end of July. Therefore, we decided to not include this site in our inversion. We also excluded some German sites with a poor data coverage (STE, JUE) to avoid an uneven impact on the inversion results during the year.

To assess the impact of distant regions with high radon fluxes on the inversion results in the Germany domain, we made a sensitivity inversion run using the additional observations from the mountain site Puy de Dôme (PUY). This site is located in the southern part of France, for which the process-based radon flux maps indicate elevated radon fluxes, too.

We initially excluded the PUY site in our inversion system due to the following reasons: The PUY site uses a LSCE radon instrument, and there has never been made an intercomparison between the LSCE and the HRM radon instruments at that specific mountain site. Therefore, it is unclear how to scale the PUY measurements to make them comparable to the HRM scale. Furthermore, the PUY site is challenging to represent in the transport model due to its very steep terrain.

Nevertheless, for this PUY inversion run, we assume a scaling factor of 0.7 to make the PUY measurements comparable to the HRM scale. This factor has been found in Schmithüsen et al. (2017) for the French site Gif-sur-Yvette (GIF), which has a similar radon intake height above ground as the PUY site. However, while the GIF site is located 167 m asl, the PUY site is located 1465 m asl, which calls into question the suitability of the assumed scaling factor. Therefore, we will only interpret the PUY inversion results in a qualitative way.

Figure 1 (below) shows the posterior flux difference between the PUY inversion, which uses the additional observations from the PUY site, and our standard inversion (without PUY). The largest flux differences occur southwest of the PUY site in the southern part of France and on the Iberian Peninsula, where our base inversion is poorly constrained due to the lack of observations. The south-westerly orientation of the footprint of the PUY site can be explained by the predominant influence of south-westerly winds in that region. Consequently, the PUY observations contain mainly information on the fluxes in south-west Europe. Indeed, the impact of the PUY observations on the fluxes in the Germany domain are very small. When including the PUY observations, the annual mean flux in the Germany domain changes by less than 3%.

From that, we expect that the potential influence from radon sites in Spain on the inversion results in the Germany domain is negligible. Those sites are even more distant than the PUY site, and the effect of dilution and radioactive decay should therefore be even larger. We added this additional study in the Appendix E of the manuscript and discuss it in the main text (l. 609-616; l. 792-822).

[Figure]

Figure 1: Posterior Rn flux difference between an inversion run with and without using the Rn observations from the PUY site (marked in yellow). Both inversions use the GLDAS prior fluxes.

**Lines 75-83:** In this paragraph authors declares that in order to avoid nocturnal PBLH associated problems the study was carried out only using 'afternoon' observations (or nocturnal at mountain sites). Behind this methodology there is the hypothesis that

diurnal/nocturnal radon fluxes are constants. However, despite the fact that the traceRadon European radon flux maps from Karstens and Levin, 2024 have a daily temporal resolution, studies have shown that radon fluxes may vary between 20%-50% during the day (Rabago et al., 2022). It should be cited as possible source of variability/uncertainty.

The selection of afternoon observations only is commonly done in inversion studies to account for the deficits in atmospheric transport models to represent stable atmospheric conditions. However, this does not directly mean that we need to assume constant radon fluxes throughout the day. As the radon half-life is almost 4 days, the afternoon-only observations contain also some information on nocturnal fluxes further away from the sites.

Here a brief example: Assume we have one afternoon radon observation made at 13 UTC at a specific site. Then we use the atmospheric transport model STILT and release 100 particles at that time and at that site and calculate their back-trajectories for 10 days. From the spatio-temporal distribution of the 100 back-trajectories we derive a footprint matrix, which shows the influence on surface fluxes emitted downwind of the site during the last 10 days. This footprint has a temporal resolution of 1 hour and shows therefore influences from diurnal and nocturnal fluxes depending on the distance to the site.

Hence, the radon inversion also constrains nocturnal fluxes despite we only use afternoon observations. However, in this study, we do not attempt to constrain the diurnal cycle of the radon flux. We only solve for daily average fluxes due to computational costs and to be consistent with the process-based (prior) radon fluxes, which are also only in daily resolution. However, we included the suggested reference in the manuscript to also highlight the sub-daily radon flux variability (l. 708-710).

**Line 84:** please add that the two process-based 222Rn flux maps were obtained within the framework of the traceRadon project (Rottger et al., 2021 ).

You are fully right; we included this information (l. 89-90).

**Methods**
The methodology in some of its section is not yet clear and well presented. First of all, I will suggest to the authors of preparing and adding at the beginning of this chapter a summary diagram of the manuscript methodology where the different inputs and outputs appear clearly. As example I am adding here one scheme presented recently by Curcoll et al., 2024.

[Figure]

**Figure 1.** Sketch of the process followed for estimating CH₄ fluxes ($j_{CH_4}$) at DEC within this study.

Thank you for this suggestion. We implemented such an overview sketch (see Fig. 2 below) in the manuscript (l. 100-102) and revised the Methods section to make it clearer and easier to follow. See our specific comments below.

[Figure]

*Figure 2: Sketch of the inversion process.*

**Line 121-123**: Authors declare that in section 3.3 they revisit this assumption (speaking about the use of soil moisture data taken from 10 cm or 40 cm depth) and then they wrote: 'A temperature dependence of the diffusion coefficient is also taken into account…'. I do not understand if this sentence was referred to the work of Karstens and Levin, 2024 or to this manuscript. I assume the first one because authors do not carry out temperature analysis. *So please rewrite this sentence.*

Yes, we only revisit the assumption of using the soil moisture data from the top 40 cm. We tried to make this clearer now (l. 131-133).

**Line 131:** Figure 1a,b does not exist, only Figure 1 does. Authors may present a new Figure 1 with the different domains used by the process-based radon flux maps, German domain and the stations locations.

We prepared a new figure, which now shows all the different domains (Fig. 2 in the manuscript).

**Line 136**: 1km$^2$ and please add also the temporal resolution of this soil moisture data

Done. It is in daily resolution (l. 146).

**Line 145:** In this line the authors declare the use of Corinne, 2018 for the pixel classification. Some comments should be done about the reliability of this inventory after 7 years and the possibility of a change of use in same grid.

Unfortunately, there is no newer version of the Corine Landcover. It is supposed to be updated every 6 years, but the status for 2024 is not set to be released until Q2/2026. According to Grunewald et al. (2021) the extent of forested areas remains more or less stable. Agricultural land is decreasing at the expense of the expansion of cities and infrastructure. More sealed areas could change radon emissions, but contribution of cities and infrastructure has not been considered in the presented work.

**Table 1.** Authors may add coloumns within this table with: 1. Radon flux map name used wihtin the manuscript, spatial and temporal resolution of these data.

We revised Tab. 1 in the manuscript and added this information (l. 163-164).

**Line 155:** The title of this subsection 2.2 is wrong. You may want to use something related with 'Atmospheric radon concentration observations'

Thank you, we changed it (l. 165).

**Line 166-169:** The slow response of the ANSTO instrument usually leads to a smoothing effect of the measured radon time series (Figures 2 and 3 of Grossi et al., 2020). This fact may affect the reproduction of fast radon changes observed wihtin the atmosphere due to wind speed and direction. Authors may comment this. In addition, the ANSTO instruments have a significant background contribution due to the accumulation of 210Po. This may leads to higher radon concentrations measurements as it was observed by authors when a ANSTO scale is applied. In a recent paper by Rottger et al., 2025 this background (intercept) has been quantified for this specific monitor but be careful because it is monitor dependent. This point maybe commented within the results discussion section.

We agree that, without correction, the slow response of radon detectors can smooth short-term atmospheric variations. However, in this study, the ANSTO data from the sites in UK are corrected using a deconvolution procedure, following the methodology described in Griffiths et al. (2016) and further validated and standardized in Kikaj et al. (2025). This

approach effectively recovers rapid variations in $^{222}$Rn activity concentrations, which are typically observed during periods of changing air mass origin, such as transitions between terrestrial and oceanic fetches or during the early morning and late evening hours when the planetary boundary layer (PBL) height changes rapidly. For the non-UK ANSTO data, we applied a first-order deconvolution by shifting the $^{222}$Rn data by one hour. Since only hourly averaged data from the afternoon period are used in this study, when atmospheric mixing is strongest and $^{222}$Rn concentrations are relatively stable, such a first-order deconvolution might be acceptable for our purposes.

It is true that ANSTO radon detectors are affected by background accumulation. However, this background increases gradually and predictably over time. Standard practice involves checking the instrumental background every 2–3 months. During these checks, the detector is isolated from ambient air and the background count rate is measured after short-lived decay products have cleared. A background trend (linear or seasonally) is built from these checks and subtracted from the raw measurements before calibration. As shown in Kikaj et al. (2025), regular background correction combined with replacing the detector head materials every five years ensures long-term measurement stability. With this approach, the impact of background on the reported radon concentrations is minimal and well controlled.

We added this information in the manuscript (l. 176-189).

**Line 179-184:** Add to the references also the work done by Grossi et al., 2016 where an analysis of the wet deposition effect was realized for an HRM too. Considering that radon data from HRM were filtered for high air humidity the authors may add a column in Table 2 declaring the number of days used from each station over the 360 days of the 2011. This will offer an information about the robustness of the dataset.

We included the reference (l. 201). We also included an additional column in Tab. 2, which indicates the number of days per year with radon data (l. 237-243).

**Figure 1**: Authors may present a new Figure 1 with the different domains used by the process-based radon flux maps, German domain and the stations locations.

We prepared a new figure (now Fig. 2 in the manuscript), which shows the different domains.

**Table 2:** Schauinsland has an inlet height of only 12 m a.g.l. HRM when the inlet is close to the surface may need a different equilibrium coefficient between radon and its progeny. Please check it in Grossi et al., 2016 and comment this problem.

As mentioned in the text, we applied an equilibrium correction for all HRM sites, which have an air intake height below 90 m. In the case of Schauinsland, we used an equilibrium correction factor of 1.25. However, we refrain from providing the equilibrium correction factor for each individual site in the manuscript, as the equilibrium-corrected HRM data are publicly available (see Fischer et al., 2024).

**Line 235:** Is the STILT simulating only 100 fictitious particles or there is an typing error? Does this affect significantly the statics of its results?

No, we only use 100 particles in STILT, as it is commonly done. In STILT the horizontal resolution of the footprint matrix is dynamically reduced as the spatial distribution of the STILT particles increases. This prevents the under-sampling of surface fluxes at times and regions where the STILT particles are distributed over extensive areas and have large gaps between each other (Gerbig et al., 2003). Therefore, this enables the usage of a small number of only 100 particles in STILT and thus reduces computational costs, while ensuring proper statistics.

We added this information in the manuscript (l. 263-267). Please see also our corresponding response to the second reviewer.

**Line 236:** No model domain is shown

We indicated the model domain in the figure (see Fig. 2 in the manuscript).

**Lines 233-244:** Authors are using 3 different models with different settings. Is it done by choice or it was not possible to harmonize them? May you please present a Table summarizing how runs where realized (N. of particles, horizontal resolution of maps, length in days of back trajectories, etc.)

The FLEXPART and NAME footprints have been calculated for another project. That's why they cover a different domain and have a different resolution than the STILT footprints. However, as the different models are not harmonised, the resulting differences in the inversion flux estimates reflect realistic variability when using different transport models and setups. Hence, these differences can be used to evaluate the robustness of the inversion-based Rn flux estimates.

We added such a table to summarize the characteristics of the different models (Tab. 3 in the manuscript, l. 306-307).

**Lines 248:** May the authors better explain why they have no information about the particles distribution? Is the residence time no calculated using the back trajectories over the domain?

Yes, the particle distribution is used to derive the footprints. However, the FLEXPART and NAME footprints have been calculated for another project. In this study, we have only access to post-processed FLEXPART and NAME footprints, which are already aggregated over time.

We included this information in the manuscript (l. 280-287).

**Lines 246-259:** This paragraph is not really clear. I will suggest to improve it.

We revised this paragraph in the manuscript to make it clearer (l. 284-304).

**Line 284:** the observation uncertainty is in the order of 10%. Why is here considered constant independently from the radon absolute values. I think, may be for future studies, taking into account the variability of the uncertainty may help to improve the model results.

In our setup, we have chosen a constant observational uncertainty of 0.5 Bq m$^{-3}$, which we think is a conservative estimate for the combined uncertainty of the Rn measurements, their calibrations and corrections, and the modelled Rn background. While we agree that the uncertainty of the Rn measurements should depend on the Rn absolute values, the uncertainty of the modelled Rn background seems to be independent of them.
To investigate the effect of using relative measurement uncertainties, we did another inversion run, where we assumed the following observational uncertainty:

$$\sigma_{obs} = \sqrt{(\sigma_{meas})^2 + (\sigma_{bg})^2} \tag{1}$$

with $\sigma_{meas}$ being 10% of the absolute Rn measurements (meas), and $\sigma_{bg}$ being 100% of the modelled Rn background (bg).

Figure 3a (below) shows for the KIT (200m) site the final uncertainties of the Rn model-data mismatch (MDM), i.e. the quadratically added uncertainties of the observations and the simulations, after having performed the so-called data-density weighting to account for correlations between model-data mismatch within one week. Using the alternative uncertainties from Eq. 1 (red curve) leads, as expected, to more variable MDM uncertainties than using the constant observational uncertainties. However, the mean MDM uncertainties of both approaches are very similar and agree within 3%.

Figure 3b shows the resulting Rn flux estimates of the inversion for both MDM uncertainties. Overall, the fluxes are very similar; the annual mean fluxes in the Germany domain agree within 2.5%. We therefore think that it is appropriate to use the constant observational uncertainties in our study.

[Figure]

*Figure 3: (a) Model-data mismatch (MDM) uncertainty after data-density weighting for the KIT (200m) site if a constant observational uncertainty of 0.5 Bq m$^{-3}$ (blue) or the alternative observational uncertainty from Eq. 1 (red) is used. (b) shows the resulting Rn flux estimates for Germany.*

**Line 287:** 0.5 Bq m-3 to 1.5 Bq m-3

Done (l. 350).

**Results**
Generally speaking, the results are interesting and well presented. However, some section may improve to gain clarity. Figures have really small size text which makes it difficult to read.

We increased the font size in some figures (e.g. in Fig. 5 in the manuscript) and tried to improve the clarity of the results section.

Regarding the analysis of the data it could be nice:
- Get rain data information from the stations may help with the analysis of the soil moisture difference between 10cm and 40cm for example

In Fig. 4 (below) we compare the 0-10 cm and 0-40 cm soil moisture averages within the Germany domain with the mean precipitation in Germany (from DWD, 2024). As expected, the soil moisture within the upper 10 cm increases first when a rain event occurs, and the soil moisture increase in the 0-40 cm is slightly delayed. The soil moisture differences between the 0-10 cm and 0-40 cm layer shows a similar pattern like the precipitation curve, illustrating that rain events lead to a higher soil moisture in the top 10 cm of the soil than in the top 40 cm of the soil. Interestingly, the DWD/GLDAS soil moisture difference between 0-10 cm and 0-40 cm is positive almost the whole year. This could be explained by the porosity, which determines the maximum soil moisture (how much water can be taken up by the soil layer) and is in the case of the DWD data higher in the top 10 cm than in the top 40 cm (GLDAS-Noah and ERA5-Land use a vertically constant porosity). During rain events, the process-based Rn fluxes tend to be smaller when they are calculated using 0-10 cm soil moisture instead of 0-40 cm soil moisture data.

We added this figure in the manuscript (Fig. G1) and refer to it in the Results and Discussions section (l. 543-545; l. 562-565; l. 655).

[Figure]

*Figure 4: (a) Mean ERA5-Land (blue), GLDAS-Noah (red) and DWD/GLDAS (yellow) soil moisture in the Germany domain, averaged between 0-10 cm (bold line) and 0-40 cm (dashed line), and mean precipitation in Germany (grey histograms). Note, that the soil moisture average corresponds to a slightly larger area (Germany domain) than the precipitation average (Germany only). (b) Difference between the soil moisture 0-10 cm and 0-40 cm averages for the Germany domain. (c) Differences between the Rn fluxes based on 0-10 cm and 0-40 cm soil moisture data.*

- Presenting a plot with a comparison of soil moisture data from GLDAS, ERA5 and DWD for some station within the German domain.

Figure 5 (below) shows for 10 ICOS ecosystem sites (data from ICOS RI, 2024) in Germany the comparison between measured and modelled soil moisture for the upper 0-10 cm soil layer. The performance of the soil moisture models varies from site to site. For the grasslands and croplands sites, the DWD and ERA5-Land models seem to describe the measured soil moisture variability better than the GLDAS-Noah model, which shows less variability. The high-resolution DWD model shows for some sites (in particular, at the grasslands site RuR and at the forest site HoH) an astonishing good agreement with the observations. However, at other sites (e.g. at the forest site Tha) it shows some biases of up to roughly 0.2 m³ m⁻³. Such differences could be explained, for example, by biases in the assumed soil porosity and in other soil cardinal values (e.g. residual moisture, wilting point, and field capacity).

Overall, this comparison illustrates the difficulty of representing local (point) measurements in a model with a limited spatial resolution. We attach this comparison plot in the Appendix H of the manuscript and refer to it in the Discussions section (l. 641-642; l. 688-690).

[Figure]

*Figure 5: Comparison between the GLDAS-Noah (red), ERA5-Land (blue) and DWD (orange) soil moisture with measurements (black) performed at different ICOS ecosystem sites in Germany. The modelled soil moisture shows the average between 0-10 cm, and the measurements show the mean soil moisture from all sensors within the 0-10 cm soil layer. The first three-letter code of the subplot titles indicates the measurement site, and the second three-letter code refers to the site class: gra – grasslands, cro – croplands, dbf – deciduous broadleaf forests, enf – evergreen needleleaf forests.*

- Having a subsection analysis some specific synoptic episode to better observe and discuss the weekly variability of the data for the Germany domain. Particularly, only if the authors are interested on it, 3-months (November 2011 – January 2022) radon flux measurements

with 3h temporal resolution are available from the traceRadon project for the PTB station (Braunschweig, Germany). I add here a plot of the data presented until now only by Grossi et al., in ICOS 2022. Radon flux time series was measured using a Autoflux instrument (Grossi et al., 2023)

[Figure]

Thank you very much for sharing the Rn flux measurements from the PTB site in Braunschweig! We plotted the different prior (flat, GLDAS, ERA5, DWD/GLDAS) and the resulting posterior flux estimates from the PTB grid cell in Fig. 6 below. On this very local scale, the different prior flux estimates range between 16.4 Bq m$^{-2}$ h$^{-1}$ and 53.0 Bq m$^{-2}$ h$^{-1}$ during the overlapping time interval (2021-11-11 until 2021-12-31). The resulting posterior fluxes are closer to each other; they range between 24.9 Bq m$^{-2}$ h$^{-1}$ and 37.9 Bq m$^{-2}$ h$^{-1}$ and show a decreasing trend between mid of November and mid of December.

Unfortunately, the flux results of our one-year inversion are less reliable in December (and January) 2021 due to edge effects. To more reliably compare our results with the PTB measurements and to make use of the full PTB flux record, we should first extend our inversion for 2022. Then, the PTB measurement period would be in the middle of the inversion period and not affected by spin-up and spin-down effects. Although we will work on extending the Rn inversion to multiple years, this is beyond the scope of the present study, which focusses on 2021. However, as soon as the multiple-year Rn inversion is ready, we are very interested in comparing the PTB measurements with our flux estimates.

[Figure]

*Figure 6: Prior (dashed) and posterior (solid) Rn flux estimates for the grid cell of the Physikalisch-Technische Bundesanstalt (PTB) site during the overlapping time period with the PTB flux measurements.*

**Figure 2**: Please add a white rectangle on each map to visualize the Germany domain.

Done (see now Fig. 3 in the manuscript).

**Figure 3**: It is not clear if the presented data represent the 'mean' values over the domains. Please clarify it.

Yes, the data in the figure show the mean values over the full EU domain (now Fig. 4c) and over the Germany domain (now Fig. 4d in the manuscript). We clarified this (l. 435-437).

**Section 3.2.2** The methodology applied to test the sensitivity of the inversion results was not previously explained in an 'ad hoc' section in the Methods. This fact makes difficult for the reader following the results presentation. Please improve it. For example Table 3 should presented and explained within the methods chapter when this sensitivity study is explained.

We added a new section (2.5) in the Methods section showing the overview of the different inversion runs. We also shifted the table into that section (is now Tab. 4 in the manuscript, l. 372-377).

**Lines 412-413** 46% and 18%, 47% and 32%

Done (l. 497-498).

**Figures 4 and 5:** Add a title with Germany Domain (2011) as for Figure 6

Done (see now Fig. 5 and 6 in the manuscript).

**Section 3.4** Title may be changed in agreement with the content

We changed the title to „Evaluating soil depth sensitivity in $^{222}$Rn flux", which was proposed by the other reviewer (l. 536).

**Figure 7:** It will be nice to add to the composite figure a plot where the 10cm-40cm difference radon fluxes and soil moisture time series may be observed for the different datasets.

We implemented such a figure in the manuscript (see Fig. G1 in the manuscript). See also our respond to your related comment further above.

**Discussion and Conclusions**
**4.1 Section:** It should be commented about the importance of not having radon data for the rest of Europe, mainly areas such as Spain, Portugal and Italy where radon fluxes may be significantly high (hundreds of Bq m-2h-1) and when, under afternoon conditions, advective winds may transport radon from these areas to Germany.

We included such a discussion in this section and added the PUY inversion study (see above) in the manuscript (l. 609-616; Appendix E).

**Line 515** add 'thanks to the traceRadon project'

Done (l. 622).

**Line 522:** add 'in comparison with STILT'

Done (l. 629).

**Line 523-524**: Here is important to explain that ANSTO instrument are affected by a cumulative background due to the 210Po and the instruments need to be calibrated at least on year basis to avoid increase in the measured radon concentration.

We added this to the Methods section of the manuscript (l. 186-189). Please also refer to our response above.

**Line 519** this sentences seems to be in disagreement with the rest of the paragraph. Is the 'no' a mistyping?

You are right, this is misleading. We revised this part in the manuscript (l. 673-678).

**Line 581-586** Some comments may be added about the influence that a temperature gradient within the soil/surface column may have.

Biases in the soil temperature seem to have a minor effect when calculating process-based $^{222}$Rn fluxes. We created an alternative ERA5-based $^{222}$Rn flux map by replacing the ERA5-Land soil temperature with the GLDAS-Noah soil temperature. In the Germany domain, the ERA5-Land and GLDAS-Noah soil temperatures of the top 40 cm of soil show a mean

difference of 1.84 ± 0.82 K in 2021. However, this difference of almost 2 K in soil temperature leads to a change of only 1% in the resulting annual mean $^{222}$Rn flux.

Moreover, no clear indications for a correlation between temperature gradient and radon flux have been found in the analysis of the traceRadon measurement campaigns. According to Duenas et al. (1997), the temperature gradient in the soil could simply be related to a vertical soil moisture gradient and not add any further information. For example, when the temperature of the top soil is high, the soil humidity is low, which in turn leads to a higher radon flux than when the temperature of the top soil is low and the humidity is high.

**Line 591:** Szegvary et al., 2009 presented a 'static' inventory was created with specific and punctual measurements. However, the creation and availability in the next future of gamma-dose rate based radon flux maps where continuous gamma dose rate time series from EURDEP may be used could be of great of interest for both climate/atmosphere and radiation protection communities. Please comment this aspect in the conclusions.

Thank you for this information. We added it in the manuscript (l. 716-727).

**Line 604**: I miss here a rough estimation of the posterior radon flux map created by the authors.

The a posteriori annual mean flux is about 14 ± 4 mBq m$^{-2}$ s$^{-1}$ (mean ± std). We added this in the manuscript (l. 741).

**Response to review 2**

The manuscript evaluates the reliability of process-based ²²²Rn emissions in Europe using atmospheric inversion modeling—an important and timely topic within the scope of Atmospheric Chemistry and Physics (ACP). By assessing different soil moisture-driven emission maps through an inversion framework constrained by ²²²Rn observations, the study provides insights into limitations of current ²²²Rn emission modeling approaches.

Major Concerns:

Writing and structure: The writing quality needs some improvement. Several section titles are overly informal, and some figure captions are too lengthy or imprecise. Enhancing the clarity, consistency, and scientific tone of the manuscript would significantly improve its accessibility and impact.

We revised our manuscript and tried to enhance the clarity and the scientific tone. We also changed several section titles to make them more precise.

Methodological clarity: There are a number of unresolved issues in the methodological approach, particularly regarding the inversion framework. These include inconsistencies in particle counts across models, the interpretation of time-aggregated versus time-resolved footprints, the background subtraction procedure, and the handling of prior–posterior dependencies. Some aspects of the inversion should be more rigorously justified or supported with sensitivity analyses.

We performed several additional sensitivity analyses to confirm our methodology and to justify our inversion results more rigorously. Please see our specific responses below.

Limited broader context: The manuscript would benefit from a broader discussion of the implications of its findings—particularly regarding how the improved ²²²Rn flux estimates might support applications beyond radon itself, such as in evaluating atmospheric transport schemes or constraining emissions of other trace gases.

We added an additional section in the Discussions part of the manuscript (see Sect. 4.4), which explains the impact of our study in a broader context.

Strengths:

Despite these concerns, the manuscript includes a number of thoughtful discussions.

The authors critically assess the data coverage of ²²²Rn observations, highlighting how observational density shapes the inversion's dependence on prior information.

The manuscript provides a breakdown of uncertainty sources, including observational scaling, model assumptions, and soil parameter variability.

The soil moisture sensitivity analysis, particularly the exploration of top-10 cm versus deeper layer effects, is a valuable contribution and has potential implications for improving future radon flux parameterizations.

Line by line comments:

Line 17: In the abstract, "In this study, we evaluate two process-based 222Rn flux maps for Europe based on two different soil moisture reanalysis products (GLDAS-Noah and ERA5-Land) using the flux results obtained from a one-year 222Rn inversion performed with the CarboScope-Regional inversion system and 222Rn observations from 17 European sites." The sentence is quite long, making it challenging to follow on a first read. My suggestion is but may not be accurate:
"This study evaluates two process-based $^{222}$Rn flux maps for Europe. The maps are developed separately using different soil moisture reanalysis products, GLDAS-Noah and ERA5-Land. The evaluation is based on flux results from a one-year $^{222}$Rn inversion, which was conducted using the CarboScope-Regional inversion system, and observational data from 17 European sites."

What is the role of the inversion system? Does it used to estimate surface 222Rn concentrations using the two soil moisture reanalysis products?

Thank you for this suggestion. We split up the long sentence to make it easier to follow. We use the inversion system to independently infer radon fluxes, using atmospheric $^{222}$Rn concentration measurements. By comparing these top-down radon flux estimates with the process-based radon flux maps, we aim to evaluate the latter. The soil moisture reanalysis products are not used in the inversion system. They are used in the process-based radon flux model to describe the diffusive transport of the $^{222}$Rn gas in the soil. We revised the abstract of the manuscript and tried to make this clearer while staying within the 250 word-limit (l. 17-21).

Line 138: "Whilst the GLDAS-Noah and ERA5-Land models assume that the soil porosity is constant over the entire soil column, AMBAV uses vertically resolved soil porosity data (from Hartmann et al., 2024)."

What could be the magnitude of the impact from using constant soil porosity (as in GLDAS-Noah and ERA5-Land) versus vertically resolved porosity (as in AMBAV)? Can this help explain the comparisons later in the manuscript?

The porosity determines the air-filled pore space, and thus directly the Rn fluxes. However, as the porosity is assumed to be temporally constant, it mainly affects the absolute Rn fluxes but not their temporal variability. To illustrate the effect of assuming vertically constant porosity, we use the DWD data product and calculate the air-filled pore space for the 0-10 cm soil layer using 0-10 cm averaged soil moisture but 0-40 cm averaged porosity (instead of 0-10 cm averaged porosity, see the black curve in Fig. 7b below). The resulting air-filled pore space is about 0.02 $m^3$ $m^{-3}$ lower than the air-filled pore space based on the 0-10 cm averaged porosity (compare black with solid orange curve in Fig. 7b).

The black curve in Fig. 7b lies between the solid orange curve (which corresponds to the air-filled pore space of the 0-10 cm soil layer based on 0-10 cm averaged DWD soil moisture and porosity data) and the solid blue curve (which corresponds to the air-filled pore space of the 0-10 cm soil layer based on 0-10 cm averaged ERA5-Land soil moisture and vertically constant porosity data). If we would use the air-filled pore space shown by the black curve in Fig. 7b to calculate Rn fluxes, the resulting fluxes would therefore also lie between the solid orange and solid blue curves in Fig. 7c. Thus, using constant instead of vertically resolved DWD porosity would reduce the resulting Rn flux by the order of roughly 2 mBq m$^{-2}$ s$^{-1}$ (10-20%).

We added this estimation to the Discussions section in the manuscript (l. 678-681).

[Figure]

Figure 7: Soil moisture (a), air-filled pore space (b) and process-based Rn fluxes (c) based on the ERA5-Land (blue), GLDAS-Noah (red), and DWD/GLDAS (orange) soil moisture data averaged over 0-10 cm (solid) and 0-40 cm (dashed) of the soil column. Note that in the case of ERA5-Land and GLDAS-Noah a vertically constant porosity is used for calculating the air-filled pore space. The DWD product provides vertically resolved porosity data; i.e., the 0-10 cm and 0-40 cm air-filled pore space is calculated using the 0-10 cm and 0-40 cm averaged porosity data, respectively. The black curve in (b) shows the air-filled pore space if the 0-10 cm averaged DWD soil moisture and the 0-40 cm averaged DWD porosity data is used, to illustrate the effect of using vertically constant porosity data.

Line ~150, Table 1: It would be better to expand Table 1 to include additional information such as temporal resolution, type of soil model used, and porosity representation (e.g., 3-hourly, constant vs. vertically resolved). This would help readers understand their influence on the $^{222}$Rn flux calculations.

Thank you for this suggestion; we revised Tab. 1 and implemented your suggestions (l. 163-164).

Line 155:Title of Section 2.2 is the same as that of Section 2.1. It could be renamed "$^{222}$Rn Observations" or something similar.

Thank you, we changed it (l. 165).

Line ~215: The caption of Table 2 is not sufficiently concise. Some information currently in the caption could be moved to the main text or directly represented in the table layout.

Latitude coordinates should explicitly indicate N (North) or S (South) to avoid ambiguity.

Consider rephrasing the caption for clarity and consistency. For example:
"Table 2. Overview of the 17 European sites providing atmospheric $^{222}$Rn measurements."
Also consider including symbol definitions as footnotes directly under the table, such as:
T – Tower sites
* – High-altitude sites

We implemented the suggested changes (l. 237-242).

Line ~227: The manuscript states that particles are released at regular hourly intervals rather than matched to the exact $^{222}$Rn measurement times. Any reason that the particles cannot be released at the exact measurement time?

You are right; this is imprecise. We changed „every hour" to "every hour of the measurements" since we use hourly-aggregated measurements (l. 251).

Line ~230: It would be better to include an equation indicating how $^{222}$Rn concentrations are calculated from the footprint and fluxes. This would clarify the convolution process and demonstrate how radioactive decay is incorporated.

We included such an equation in the manuscript (l. 254-258).

Line 235–240:The manuscript notes that 100 particles are released in STILT, compared to 20,000 in FLEXPART. This discrepancy raises concern. After 10 days of backward transport, especially within the boundary layer, the number of STILT particles in the footprint layer may be very small, potentially causing biased flux estimates. Could the authors clarify why such a low number is used in STILT? would increasing the number of STILT particles (e.g., to match FLEXPART) improve consistency and reduce sampling error?

In STILT the horizontal resolution of the footprint matrix is dynamically reduced as the spatial distribution of the STILT particles increases. This prevents the under-sampling of surface fluxes at times and regions where the STILT particles are distributed over extensive areas and have large gaps between each other (Gerbig et al., 2003). Therefore, this enables the usage of a small number of only 100 particles in STILT and thus reduces computational costs, while ensuring proper statistics.

To demonstrate that this method works well, we modelled the Rn concentrations at the KIT site using the 5-fold number of particles (i.e. 500) in STILT. Figure 8a (below) shows the Rn concentration difference between the STILT runs with 500 and 100 particles. The mean value is -0.006 Bq m$^{-3}$, which is about 0.2% of the (modelled) annual mean Rn concentration at the KIT site. Therefore, we do not expect any bias due to the fact that we used only 100 particles in STILT.

Figure 8b (below) shows the Rn concentration difference between two separate STILT runs, each based on 100 particles. These differences are caused by the random process involved in the STILT back-trajectory calculation. Their standard deviation (0.61 Bq m$^{-3}$) is similar to the standard deviation of the differences between the 500 and 100 particle STILT simulations (0.47 Bq m$^{-3}$), indicating that the variability in the difference between the 500 and 100 particle runs can be explained by the random process alone.

We added the first paragraph of our answer in the manuscript (l. 263-267).

[Figure]

Figure 8: (a) Radon activity concentration difference between a 500-particles and a 100-particles STILT run for the KIT (200m) site. (b) Radon activity concentration difference between two separate 100-particles STILT runs for the KIT site. The difference is caused by the random process in STILT.

Line 235–245: The authors use 30-day back-trajectories for FLEXPART and NAME, but only 10 days for STILT. Given the $^{222}$Rn half-life of 3.8 days, the contribution from emissions beyond ~10 days should be minimal. Could the authors clarify whether the 30-day simulations were necessary for resolving $^{222}$Rn fluxes?

The FLEXPART and NAME runs were initially done for another project and another, i.e. much larger, model domain. Therefore, the back-trajectories were calculated for 30 days, to ensure that most of the particles have left this larger model domain. However, to resolve Rn fluxes in Europe, a 10-day simulation, as is done with STILT, is sufficient. We added this information in the manuscript (l. 280-282).

Line ~245–255: FLEXPART and NAME use $^{222}$Rn-decay-corrected footprint products aggregated over 30 days, and that these footprints lack temporal resolution. Without knowing when the footprints are most sensitive, it is unclear how the daily fluxes can be meaningfully convolved with the aggregated footprints.

To investigate for which Rn flux times the FLEXPART and NAME footprints are most sensitive, we performed the experiment with the time-aggregated STILT footprints (see l. 284-304 in the manuscript, and our answer to your next comment below).

Line ~255–260: The authors justify their use of 3-day averaged fluxes with FLEXPART and NAME based on timing sensitivity inferred from STILT. However, STILT uses only 100 particles, which may not be sufficient to robustly resolve the timing distribution after 3–10 days of transport. Moreover, FLEXPART is capable of outputting time-resolved footprint information. There are critical compromises in the intercomparison of the three model products.

The FLEXPART and NAME footprints have been calculated for another project. For this study we have only post-processed FLEXPART and NAME footprints, which are already aggregated over time. That's why we performed the timing sensitivity experiment with the STILT footprints. Due to the dynamical adjustment of the horizontal resolution of the STILT footprints, 100 particles are sufficient in STILT for enabling proper statistics; i.e., there is no benefit in increasing the STILT particle number (see Fig. 8 and our answer above). Therefore, we argue that the STILT model is sufficiently similar to the FLEXPART and NAME models to be used for determining their timing sensitivity.

We revised the corresponding section in the manuscript (l. 284-304).

Line ~260–275: The manuscript applies a modeled $^{222}$Rn background correction by subtracting TM3-simulated concentrations based on a globally uniform $^{222}$Rn flux map. However, this approach raises several concerns. First, the short lifetime of $^{222}$Rn means that long-range contributions to observed concentrations are inherently small. Subtracting a modeled background may introduce biases and artificially reduce the observational constraint. I do not think this is necessary. Second, the assumed constant fluxes outside Europe lack empirical justification and do not account for known variability in soil radium content, land cover, or meteorology. This can introduce noise in the seasonal/spatial variations.

We agree that inaccuracies in the simulated background directly lead to biases in the flux estimates of the regional inversion. To investigate the impact of the simulated background on the Rn flux results, we performed two additional inversion runs using alternative Rn backgrounds. For the first run, we replaced the TM3-simulated Rn concentration field by the Rn concentration field provided by the Copernicus Atmosphere Monitoring Service (CAMS) global reanalysis (EAC4, Inness et al., 2019) and again used the endpoints of the STILT trajectories to calculate an alternative (decay-corrected) Rn background. If averaged over all sites, the CAMS-based Rn concentration field leads to almost 50% lower annual mean Rn background concentrations than the TM3-based Rn concentration field (0.14 Bq m$^{-3}$ vs. 0.26 Bq m$^{-3}$). For the second run, we fully neglected the Rn background, i.e. we assumed that it is zero.

Figure 9 (below) compares the results of these additional two inversion runs with our standard inversion run based on the TM3-simulated Rn background. A smaller Rn background yields higher Rn concentrations used in the inversion, and thus higher Rn fluxes. The inversion run based on a zero Rn background leads to 12% higher fluxes than our standard inversion based on the TM3-simulated background. The CAMS-based Rn

background leads to 6% higher fluxes than the TM3-simulated background. The temporal variability of the Rn fluxes is less affected by the choice of the Rn background.

Overall, the range between the 3 curves in Fig. 9 (below) indicates the uncertainty in the fluxes due to the Rn background. From that we conclude that the TM3-simulated Rn background seems to be an upper estimate, whereas a zero Rn background might lead to slightly overestimated Rn fluxes.

We added this analysis in the manuscript (see Fig. F1 and l. 323-327; Tab. 4; Fig. 5; l. 630-633).

[Figure]

*Figure 9: Impact of the European Rn background on the Rn flux estimates for the Germany domain. Shown are the results of inversions using a TM3-simulated background (solid grey curve), a CAMS (EAC4)-based background (orange), and a zero background (magenta).*

Line ~280–285 (in the inversion system description): Including an equation of the inversion system would clarify how the posterior fluxes are derived and how prior information and observational constraints are weighted. A simple expression such as

$$J(x) = (x - x_{\text{prior}})^T \mathbf{B}^{-1}(x - x_{\text{prior}}) + (y - Hx)^T \mathbf{R}^{-1}(y - Hx)$$

along with brief definitions of terms.
x: unknown posterior $^{222}$Rn flux field; x prior: prior flux map
y: observed $^{222}$Rn concentrations; H: footprint-convolved fluxes

This may raise another critical issue. If my expression is correct, then H is NOT independent from x prior. It is calculated based on the flux map, which is x prior here. I am not sure if the inversion model is still valid if H is a dependent of x prior. Again, my expression can be wrong, so it is important for the authors to provide such an expression.

In the cost function that you provided, H is the atmospheric transport matrix, which incorporates the surface sensitivities of all sites (i.e. the footprints). H itself is independent of the prior fluxes! Convolving the footprints with the fluxes yields the modelled concentrations, Hx, which are then subtracted from the observations y. By minimizing the cost function J(x), the inversion iteratively reduces this model-data mismatch.

In CarboScope, the Bayesian cost function is technically implemented in a slightly different way by using a "linear statistical flux model" (Rödenbeck et al., 2009):

$$x = x_{fix} + Fp \tag{2}$$

The flux field x is written as the sum of a fixed flux component $x_{fix}$, which is the mean of the prior flux field $x_{prior}$, and the product between a matrix F and a vector p with adjustable parameters. By construction, the a priori parameters in p have zero mean, unit variance, and are uncorrelated. The columns of F represent spatio-temporal flux patterns, which are scaled by the elements of p. The prior covariance matrix B is then given by $B = FF^T$ and the cost function J(p) is:

$$J(p) = \frac{1}{2}p^T p + \frac{1}{2}(y - Hx)^T R^{-1}(y - Hx) \tag{3}$$

The second term in Eq. 3 shows the data constraint and the first term in Eq. 2 describes the prior constraint.

We added these equations in the manuscript (l. 334-345).

Line ~310, Section Title: The title "Temporal correlation between soil moisture and 222Rn flux variability" sounds like it will present a data-driven correlation study between two independently measured quantities: soil moisture and [222]Rn flux variability. In fact, it investigates the impact of using a flat prior compared to the soil moisture based prior. I believe that this involves not only temporal variations but also spatial variations. The title can be something like "2.4 The impact of using Soil Moisture based [222]Rn fluxes compared to a flat prior on the inversion system". This is probably not good enough. I leave it here for the authors to polish more.

We revised this section by also taking into account a comment from the first reviewer. Its title is now "2.5 Overview of the different inversion runs" (l. 372).

Line ~360, Figure 3: The maps present annual means. However, given the high temporal variability of [222]Rn fluxes driven by soil moisture and meteorology (as shown in the time series), the maps cannot provide much information. It would strengthen the evaluation to also show the innovations at monthly or seasonal resolution.

We prepared an additional figure showing for each quarter of 2021, the 3-monthly averaged innovation for the inversion runs based on the GLDAS and ERA5 prior fluxes, respectively (see Fig. 10 below). While in the case of the GLDAS inversion there are also some negative flux adjustments in France during summer, the ERA5 inversion shows positive flux adjustments in France and Germany throughout the whole year.

We added this figure in the manuscript and refer to it when describing the results (Fig. D1; l. 411-412; 416-418).

[Figure]

*Figure 10: Innovation for CSR-STILT inversion runs using the GLDAS (red) and ERA5 (blue) prior Rn fluxes, averaged for the first (a), second (b), third (c), and forth (d) quarter of 2021.*

Line ~370: Figure 3 shows that posterior $^{222}$Rn fluxes over the European domain exhibit a clear seasonal cycle, while the Germany-only domain shows much weaker seasonal variability, particularly in the flat-prior inversion. It would be better if the authors can discuss this difference and its implications for interpreting regional versus continental-scale flux patterns.

The flux estimates for the whole European domain only show a substantial seasonal cycle when the process-based prior fluxes are used, which in turn also exhibit a pronounced seasonal cycle. Using the flat prior results in much weaker seasonal variability in the posterior fluxes. This demonstrates that the continental-scale inversion results and their seasonal variability are heavily influenced by the prior information. Thus, our data coverage is insufficient to reliably constrain the seasonal variability of the European Rn fluxes.

Therefore, we refrain from drawing conclusions about the seasonal cycle of the continental-scale Rn fluxes, and instead focus our analysis on the Germany domain, which is well covered by observations.

In the Germany domain, the different inversion runs produce similar seasonal variability, demonstrating that this region is indeed well constrained by observations, and that the inversion is less dependent on prior information. In the process-based prior fluxes, the seasonal variability is mainly driven by soil moisture. In the Germany domain, the GLDAS and ERA5 prior fluxes are 29% and 37% higher in the summer half-year of 2021 than in the winter half-year. The corresponding posterior fluxes show 36% and 27%, respectively, higher fluxes in the summer half-year compared to the winter half-year. The flat-prior inversion leads to a very similar result: the flux estimates in the summer half-year are 30% higher than in the winter half-year.

We included this in the manuscript (l. 422-428; 443-447).

Line ~378: A more precise title could be "Sensitivity of Inversion Results to Model Configuration"

We adopted your suggestion (l. 461).

Line ~402: "The NAME forward simulations" have never been introduced before this point.

The „NAME forward simulation" means the mapping of the Rn prior fluxes with the NAME footprints. We clarified this (487-489).

Line ~417: "However, there seems to be a slight seasonal cycle in the difference". Again, it would be better to have some discussion about reasons for such seasonal cycles.

The small seasonal cycle in the differences could indicate on seasonal variations in the model performances. This should be investigated further, e.g. by checking if it also occurs in other years. We added this in the manuscript (l. 504-506).

Line ~448: The title "Alternative $^{222}$Rn flux maps" does not clearly reflect the scope of the section, which focuses on analyzing the temporal variability of posterior $^{222}$Rn fluxes and their correlation with soil moisture at different depths. Try from "Evaluating Soil Depth Sensitivity in $^{222}$Rn Flux".

We adopted your suggestion (l. 536).

Line ~475: The authors state that the annual mean flux is only marginally affected (2–3%). It would be valuable to also assess the impact on daily or monthly mean fluxes. Such resolution is particularly relevant for interpreting short-term atmospheric $^{222}$Rn observations and improving result fidelity.

The daily $^{222}$Rn fluxes can differ by up to roughly 30% (ERA5) and 60% (GLDAS) if the 10 cm soil moisture is used instead of the 40 cm soil moisture. We added this information in the manuscript (l. 565-566).

Line ~516: Although I had a few concerns about the methodology, the discussions here are quite thoughtful. The sources of the big difference between process-based fluxes and the posterior fluxes are clearly articulated, and the uncertainty due to model assumptions versus prior biases is discussed carefully.

Thank you for your feedback.

[revised manuscript text omitted]